# ACUS: AUDIO CAPTIONING WITH UNBIASED SLICED WASSERSTEIN KERNEL

## ABSTRACT

Teacher-forcing training for audio captioning usually leads to exposure bias due to training and inference mismatch. Prior works propose the contrastive method to deal with caption degeneration. However, the contrastive method ignores the temporal information when measuring similarity across acoustic and linguistic modalities, leading to inferior performance. In this work, we develop the temporal-similarity score by introducing the unbiased sliced Wasserstein RBF (USW-RBF) kernel equipped with rotary positional embedding to account for temporal information across modalities. In contrast to the conventional sliced Wasserstein RBF kernel, we can form an unbiased estimation of USW-RBF kernel via Monte Carlo estimation. Therefore, it is well-suited to stochastic gradient optimization algorithms, and its approximation error decreases at a parametric rate of $\mathcal{O}(L^{-1/2})$ with $L$ Monte Carlo samples. Additionally, we introduce an audio captioning framework based on the unbiased sliced Wasserstein kernel, incorporating stochastic decoding methods to mitigate caption degeneration during the generation process. We conduct extensive quantitative and qualitative experiments on two datasets, AudioCaps and Clotho, to illustrate the capability of generating high-quality audio captions. Experimental results show that our framework is able to increase caption length, lexical diversity, and text-to-audio self-retrieval accuracy. We also carry out an experiment on two popular encoder-decoder audio captioning backbones to illustrate that our framework can be compatible with a diversity of encoder-decoder architectures.

## 1 INTRODUCTION

Audio captioning task (Drossos et al., 2017) strives to describe acoustic events and their temporal relationship in natural language. Compared to other audio-related tasks, audio captioning is a multimodal learning task which lies at the intersection of audio and natural language processing. There are two common architectures for audio captioning: encoder-decoder (Kim et al., 2024; Mei et al., 2024) and prefix-tuning (Deshmukh et al., 2023; Kim et al., 2023) architectures. The former architecture consists of an audio encoder and a language model as a text decoder, and both the encoder and decoder are trained at the training phase. On the other hand, the former architecture has a pretrained language model and a trainable audio encoder which is finetuned during the training phase. The popular framework for audio captioning is to train audio captioning models by maximizing the likelihood of ground-truth captions during the training stage and then utilizing trained models to generate audio captions at the inference stage.

Although audio captioning models trained with maximum likelihood procedures are capable of generating plausible audio captions, they still suffer from exposure bias due to training and inference mismatch. (Schmidt, 2019) conducted a comprehensive study regarding exposure bias and argues that exposure bias can be viewed as a generalization issue for language models trained by teacher forcing procedures. Therefore, regularization techniques (Shi et al., 2018; An et al., 2022) are proposed to alleviate exposure bias in language models. (An et al., 2022) proposed a contrastive loss regularization for conditional text generation. The contrastive loss is jointly optimized with likelihood loss to mitigate exposure bias for language models. Then, the prediction sequence is chosen by maximizing the likelihood and cosine similarity between a prefix-text and generated sequences. The contrastive method is efficient for conditional text generation, but it is not well-suited for the

audio captioning task. The cosine similarity induced by contrastive loss is unable to consider temporal information between audio and caption sequences when measuring the similarity between them. Thus, the cosine similarity is inadequate to rerank candidate captions at the inference stage.

Dynamic Time Warping (DTW) (Sakoe & Chiba, 1978) and Soft Dynamic Time Warping (soft-DTW) (Cuturi & Blondel, 2017) are two widely adopted distances used to measure the discrepancy between two time series. They are capable of considering temporal information, however, the monotonic alignment imposed by DTW is too strict and might adversely affect the measurement of the discrepancy between audio and caption when local temporal distortion exists. (Su & Hua, 2017) proposed an order-preserving Wasserstein distance to deal with the shortcoming of DTW. Although the order-preserving Wasserstein distance can measure the discrepancy between two sequential data when temporal distortion exists, it is ineffective to measure the discrepancy between high-dimensional sequences due to the dimensionality curse of the Wasserstein distance.

To address all aforementioned issues, we propose the Audio Captioing with Unbiased sliced Wasserstein kernel (ACUS) framework to alleviate the caption degeneration for the audio captioning task and better measure cross-modal similarity. We develop the unbiased sliced Wasserstein RBF kernel (USW-RBF) for precisely measuring the similarity score between acoustic and linguistic modalities. The USW-RBF leverages the radial basis function (RBF) kernel, in which the sliced Wasserstein distance equipped with the rotary positional embedding is used as the distance. The proposed kernel is unbiased. Hence, it is highly compatible with stochastic gradient optimization algorithms, and its approximation error decreases at a parametric rate of $\mathcal{O}(L^{-1/2})$. We also derive the proposed kernel and show that it is capable of measuring the similarity in terms of features and temporal information. Furthermore, (Arora et al., 2022a) provides an analysis of exposure bias through the lens of imitation learning and empirically shows that stochastic decoding methods are able to alleviate exposure bias for language models. According to this observation, we leverage the ACUS framework with stochastic decoding methods at the inference stage to rerank generated captions to choose the most suitable candidate caption. To sum up, our contributions can be summarized as follows:

1. We propose the USW-RBF kernel to precisely measure the similarity between acoustic and linguistic modalities for encoder-decoder audio captioning models. Our kernel is able to deal with the dimensionality curse and temporal distortion by leveraging the sliced Wasserstein distance equipped with rotary positional embedding.

2. We analyze the USW-RBF kernel and prove that it is an unbiased kernel. Thus, it is well-suited to stochastic gradient optimization algorithms, with its approximation error diminishing at a parametric rate of $\mathcal{O}(L^{-1/2})$ with $L$ Monte Carlo samples.

3. We propose the ACUS framework which leverage stochastic decoding methods, such as nucleus and top-k samplings, at the inference stage to significantly alleviate exposure bias for the audio captioning task.

## 2 BACKGROUND

### 2.1 ENCODER-DECODER AUDIO CAPTIONING

An encoder-decoder audio captioning model, denoted as $\mathcal{M} = (f_\theta, g_\phi)$, is capable of generating captions $\mathbf{y} = \{y_t\}_{t=0}^N$ conditioning on a given audio $\mathbf{x}$. Here, $f_\theta$ ($\theta \in \Theta$) and $g_\phi$ ($\phi \in \Phi$) are the encoder and decoder parameterized by $\theta$ and $\phi$ respectively. The encoder is designed to extract acoustic features from audio, while the decoder is able to decode extracted acoustic features to natural language. The audio captioning model is trained to maximize the likelihood of ground-truth captions when predicting the current word in the sequence given the prior words $y_{<t}$ and the hidden representation of audio $z_x = f_\theta(x)$. The training objective for the audio captioning model is defined as follows:

$$\mathcal{L}_{MLE} = -\sum_{t=1}^N \log p_{g_\phi}(y_t | z_x, y_{<t}). \tag{1}$$

After training, the pretrained encoder-decoder model $\mathcal{M}$ is utilized to generate the most explainable caption for a given audio. Typically, beam search decoding is used to generate $\mathcal{B}$ candidate captions,

and then the caption with the highest probability is chosen as the prediction

$$\hat{\mathbf{y}} = \underset{\mathbf{y_i} \in \mathcal{B}}{\arg\max}\, p_{g_\phi}(\mathbf{y_i}|z_x).$$

(2)

There is a critical issue with likelihood training, which is exposure bias. The audio captioning model predicts the next word based on previous ground-truth words $y_{<t} \in y$ at the training stage, but it adopts the predicted tokens $\hat{y}_{<t}$ by itself to generate the next token $\hat{y}_t$ at inference stage. Due to exposure bias, there is a significant gap in terms of performance of pretrained audio captioning models on training and test data. Furthermore, the beam search decoding even makes the exposure bias more critical due to error accumulation.

## 2.2 CONTRASTIVE LEARNING FOR AUDIO CAPTIONING

To mitigate the exposure bias with likelihood training, contrastive learning for audio captioning (Chen et al., 2022a; Liu et al., 2021) introduces a contrastive objective which aims to maximize cosine similarity between audio and ground-truth caption. Negative examples are directly drawn from minibatch as follows SimCLR (Chen et al., 2020) to compute the infoNCE loss (Oord et al., 2018)

$$\mathcal{L}_{NCE} = -\log \frac{\exp(\cos(z_x, z_y)/\tau)}{\sum_{y' \in Y} \exp(\cos(z_x, z_{y'})/\tau)},$$

(3)

where $z_x, z_y, z_{y'} \in \mathbb{R}^d$ denote the hidden representation of audio input $x$, ground-truth caption $y$, and caption $y' \in Y$ from the minibatch, respectively. The temperature $\tau > 0$ is utilized to control the strength of penalties on negative examples. The likelihood objective is jointly optimized with the contrastive loss at the training phase

$$\mathcal{L} = \mathcal{L}_{MLE} + \mathcal{L}_{NCE}.$$

(4)

There are two benefits of contrastive regularization: (1) alleviating exposure bias by regularizing audio and caption hidden representations and (2) leveraging the cosine similarity function between audio and ground-truth caption hidden representations learned during training for reranking generated captions. Denote $\mathcal{B}$ as generated captions using decoding methods such as beam search or nucleus sampling (Holtzman et al., 2020), the corresponding caption for the given audio $x$ is chosen as

$$\hat{\mathbf{y}} = \underset{\mathbf{y_i} \in \mathcal{B}}{\arg\max}\{p_{g_\theta}(\mathbf{y_i}|z_x) + \cos(z_x, z_{y_i})\}.$$

(5)

Although contrastive regularization is effective in mitigating exposure bias for audio captioning, the similarity between audio and ground-truth caption hidden representation is computed based on cosine similarity between the average pooling of audio and caption hidden representation. The average pooling operation discards the temporal information in audio and caption representation, therefore, leveraging contrastive regularization for inference can lead to inferior performance.

## 3 METHODOLOGY

We first develop the unbiased sliced Wasserstein RBF kernel (USW-RBF) to deal with the dimensionality curse and strict monotonic alignment for measuring similarity across multimodalities. The USW-RBF is equipped with the rotary positional embedding to consider temporal information when measuring similarity across linguistic and acoustic modalities. Then, we propose the Audio Captioing with Unbiased sliced Wasserstein kernel (ACUS) framework to mitigate text degeneration for audio captioning. We leverage stochastic decoding methods with the USW-RBF as similarity score across modality to alleviate exposure bias at the inference stage. Our training and inference procedure are illustrated in Figure 1.

### 3.1 UNBIASED SLICED WASSERSTEIN KERNEL

**Wasserstein distance.** Given $p \geq 1$, Wasserstein distance (Peyré et al., 2019) between $\mu$ and $\nu$ be two distributions belongs to $\mathcal{P}_p(\mathbb{R}^d)$ is defined as:

$$W_p^p(\mu, \nu) = \inf_{\pi \in \Pi(\mu, \nu)} \int_{\mathbb{R}^d \times \mathbb{R}^d} \|x - y\|^p d\pi(x, y)$$

where $\Pi(\mu, \nu)$ is the set of all distributions that has the first marginal is $\mu$ and the second marginal is $\nu$ i.e., transportation plans or couplings.

**Sliced Wasserstein distance.** Given $p \geq 1$, the sliced Wasserstein (SW) distance Bonneel et al. (2015); Nguyen et al. (2021); Nguyen & Ho (2024) between two probability distributions $\mu \in \mathcal{P}_p(\mathbb{R}^d)$ and $\nu \in \mathcal{P}_p(\mathbb{R}^d)$ is defined as:

$$SW_p^p(\mu, \nu) = \mathbb{E}_{\psi \sim \mathcal{U}(\mathbb{S}^{d-1})}[\mathrm{W}_p^p(\psi\sharp\mu, \psi\sharp\nu)], \tag{6}$$

where the one dimensional Wasserstein distance has a closed form which is:

$$\mathrm{W}_p^p(\psi\sharp\mu, \psi\sharp\nu) = \int_0^1 |F_{\psi\sharp\mu}^{-1}(z) - F_{\psi\sharp\nu}^{-1}(z)|^p dz$$

where $F_{\psi\sharp\mu}$ and $F_{\psi\sharp\nu}$ are the cumulative distribution function (CDF) of $\psi\sharp\mu$ and $\psi\sharp\nu$ respectively. When $\mu$ and $\nu$ are empirical distributions over sets $Z_x = \{z_x^1, \ldots, z_x^N\}$ and $Z_y = \{z_y^1, \ldots, z_y^M\}$ i.e., $\mu = \frac{1}{N} \sum_{i=1}^N \delta_{z_x^i}$ and $\nu = \frac{1}{M} \sum_{j=0}^M \delta_{z_y^j}$ respectively, $\psi\sharp\mu$ and $\psi\sharp\nu$ are empirical distributions over sets $\psi^\top Z_x = \{\psi^\top z_x^1, \ldots, \psi^\top z_x^N\}$ and $\psi^\top Z_y = \{\psi^\top z_y^1, \ldots, \psi^\top z_y^M\}$ in turn (by abusing the notation of matrix multiplication). As a result, the quantile functions can be approximated efficiently.

**Monte Carlo estimation of SW.** In practice, the sliced Wasserstein is computed by the Monte Carlo method using $L$ samples $\psi_1, \ldots, \psi_L$ sampled from the uniform distribution on the unit sphere $\mathcal{U}(\mathbb{S}^{d-1})$ due to the intractability of the expectation:

$$\widehat{SW}_p^p(\mu, \nu; L) = \frac{1}{L} \sum_{l=1}^L W_p^p(\psi_l\sharp\mu, \psi_l\sharp\nu), \tag{7}$$

where $L$ is referred to as the number of projections. When two empirical distributions have the same number of supports i.e., $\mu = \frac{1}{N} \sum_{i=1}^N \delta_{z_x^i}$ and $\nu = \frac{1}{M} \sum_{j=0}^N \delta_{z_y^j}$, we have:

$$\widehat{SW}_p^p(\mu, \nu; L) = \frac{1}{L} \frac{1}{N} \sum_{l=1}^L \sum_{i=1}^N \|\psi^\top z_x^{\sigma_{1,l}(i)} - \psi^\top z_y^{\sigma_{2,l}(i)}\|_p^p,$$

where $\sigma_{1,l} : [[N]] \to [[N]]$ and $\sigma_{2,l} : [[N]] \to [[N]]$ are two sorted permutation mapping of $\psi^\top Z_x$ and $\psi^\top Z_y$ in turn. By abusing of notation, we will use the notation $\widehat{SW}_p^p(Z_x, Z_y; L)$ later when $\mu$ and $\nu$ are empirical distributions over $Z_x$ and $Z_y$.

**Sliced Wasserstein RBF kernels.** Given the definition of SW in Equation (6), we can define the sliced Wasserstein RBF (SW-RBF) kernel (Carriere et al., 2017; Kolouri et al., 2016) as:

$$\mathcal{K}_\gamma(\mu, \nu) = \exp\left(-\gamma SW_p^p(\mu, \nu)\right), \tag{8}$$

where $\gamma > 0$ is the bandwidth. The $\mathcal{K}_\gamma(\cdot, \cdot)$ is proven to be positive definite (Kolouri et al., 2016) for absolute continuous distributions. The SW-RBF is intractable due to the intractability of the SW. In practice, SW-RBF is estimated by plugging in the Monte Carlo estimation of SW. However, the resulting estimation $\widehat{\mathcal{K}}_\gamma(\mu, \nu) = \exp\left(-\gamma \widehat{SW}_p^p(\mu, \nu)\right)$ is biased since the expectation is inside the exponential function.

**Unbiased Sliced Wasserstein RBF kernel.** To address the unbiasedness problem of the SW kernel, we propose a new kernel:

**Definition 1** *Given two probability distributions $\mu, \nu \in \mathcal{P}(\mathbb{R}^d)$, $\kappa \in \mathbb{R}_+$, $p \geq 1$, the unbiased sliced Wasserstein RBF kernel (USW-RBF) is defined as:*

$$\mathcal{U}\mathcal{K}_\gamma(\mu, \nu; p) = \mathbb{E}_{\psi \sim \mathcal{U}(\mathbb{S})^{d-1}}\left[\exp\left(-\gamma W_p^p(\psi\sharp\mu, \psi\sharp\nu)\right)\right]. \tag{9}$$

**Proposition 1** *The USW-RBF kernel with $p = 2$ is a positive definite kernel for all $\gamma > 0$ and absolute continuous probability distributions $\mu$ and $\nu$.*

Proof of Proposition 1 is given in Appendix A.1.1. Since the USW-RBF kernel is positive definite, it is equivalent to a reproducing kernel Hilbert space and celebrates the representer theorem.

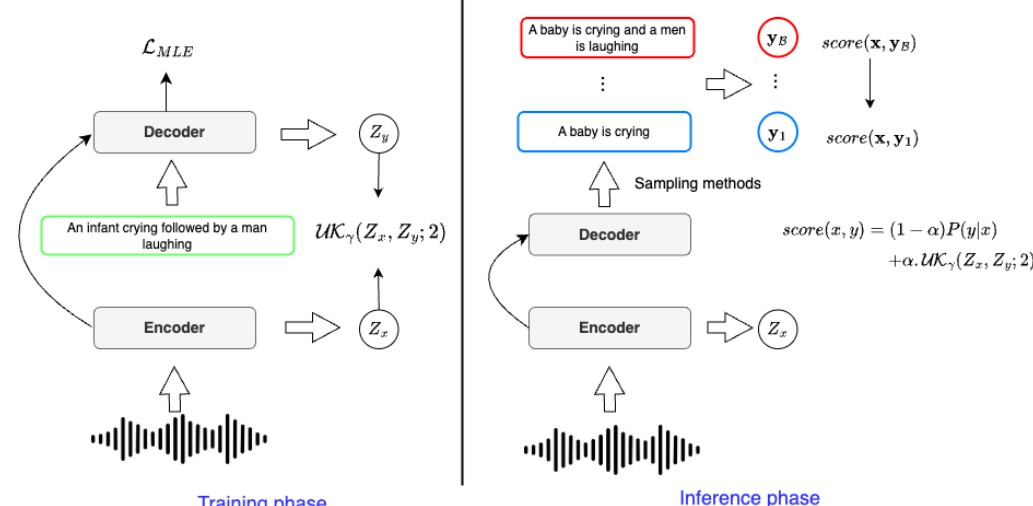

Figure 1: An overview of training and inference stage of the ACUS framework. $Z_x$ and $Z_y$ are two sequential latent representations of audio and caption, respectively.

**Proposition 2** *The USW-RBF kernel is an upper-bound of the SW-RBF kernel.*

Proposition 2 comes directly from the Jensen inequality, however, we provide the proof in Appendix A.1.2 for completeness.

Let $\psi_1, \ldots, \psi_L \overset{i.i.d}{\sim} \mathcal{U}(\mathbb{S}^{d-1})$, the USW-RBF kernel can be estimated as:

$$\widehat{\mathcal{UK}}_\gamma(\mu, \nu; p, L) = \frac{1}{L} \sum_{l=1}^{L} \exp\left(-\gamma W_p^p(\psi_l \sharp \mu, \psi_l \sharp \nu)\right). \tag{10}$$

It is worth noting that Quasi-Monte Carlo methods (Nguyen et al., 2024) and control variates techniques (Nguyen & Ho, 2023; Leluc et al., 2024) can also be applied to achieve more accurate approximation. However, we use the basic Monte Carlo to make theoretical investigation easier.

**Proposition 3** *Given $\psi_1, \ldots, \psi_L \overset{i.i.d}{\sim} \mathcal{U}(\mathbb{S}^{d-1})$, $p > 1$, and $\mu, \nu \in \mathcal{P}(\mathbb{R}^d)$ ($d \geq 1$), we have:*

*(i) $\widehat{\mathcal{UK}}_\gamma(\mu, \nu; p, L)$ is an unbiased estimate of $\mathcal{UK}_\gamma(\mu, \nu)$ i.e., $\mathbb{E}[\widehat{\mathcal{UK}}_\gamma(\mu, \nu; p, L)] = \mathcal{UK}_\gamma(\mu, \nu; p)$,*

*(ii) $\mathbb{E}\left|\widehat{\mathcal{UK}}_\gamma(\mu, \nu; p, L) - \mathcal{UK}_\gamma(\mu, \nu; p, L)\right| \leq \frac{1}{\sqrt{L}} Var\left[\exp\left(\gamma W_p^p(\psi \sharp \mu, \psi \sharp \nu)\right)\right].$*

The proof of Proposition 3 is given in Appendix A.1.3. The unbiasedness (i) is crucial for the convergence of stochastic gradient algorithms which optimizes the kernel as a loss. The bound in (ii) suggests that the approximation error decreases at a parametric rate of $\mathcal{O}(L^{-1/2})$.

## 3.2 AUDIO CAPTIONING WITH THE UNBIASED SW-RBF KERNEL FRAMEWORK

**Positional encoding for USW-RBF kernel**. Given a pair of audio and ground-truth caption is denoted as $(x, y)$, the hidden representation of audio outputs by the encoder denoted as $Z_x = [z_x^1, ..., z_x^N]$, where $z_x^i \in \mathbb{R}^d$, and the hidden representation of ground truth caption conditioning on the audio outputs by the decoder denoted as $Z_y = [z_y^1, ..., z_y^M]$ where $z_y^j \in \mathbb{R}^d$. Although the USW-RBF is effective in measuring the similarity between two sets of vectors, the order of vectors within a set is not taken into account when computing the sliced Wasserstein distance. More importantly, the order of vectors within a set contains the temporal information between them, which is crucial for audio and language modality. To preserve the temporal information, we define the temporal-information preserving vector as follows

$$\phi_x^n = concat(z_x^n, pos(n)) \tag{11}$$

where $n$-th denotes the position of vector $z_x^n \in \mathbb{R}^d$ in a sequence of vector $Z_x \in \mathbb{R}^{N \times d}$, and $pos(n) \in \mathbb{R}^k$ is the corresponding positional embedding vector. there are two popular positional embedding functions: absolute positional embedding Vaswani et al. (2017) and rotary positional embedding functions (Su et al., 2024). We redefine $Z_x = [\phi_x^1, \ldots, \phi_x^N]$ and $Z_y = [\phi_y^1, \ldots, \phi_y^M]$ respectively.

**Training with the USW-RBF kernel.** We assume that $N = M$, two projected-one dimensional sequences $a_\psi = [a_1, ..., a_N]$ and $b_\psi = [b_1, ..., b_N]$, where $a_i = \psi^\top \phi_x^i$ and $b_j = \psi^\top \phi_y^j$. We denote the $\sigma_1 : [[N]] \to [[N]]$ and $\sigma_2 : [[N]] \to [[N]]$ as two sorted permutation mapping of $a_\psi$ and $b_\psi$ in turn. Let denote the projection vector $\psi = concat(\psi_1, \psi_2)$ is the concatenation of two vectors $\psi_1 \in \mathbb{R}^d$ and $\psi_2 \in \mathbb{R}^k$. Now, we define the temporal-similarity score based USW-RBF with $p = 2$:

$$
\begin{aligned}
\mathcal{UK}_\gamma(Z_x, Z_y; 2) &= \mathbb{E}_{\psi \sim \mathcal{U}(\mathbb{S}^{d+k-1})} \left[ \exp\left( -\gamma \sum_{i=1}^N (a_{\sigma_{\psi,1}(i)} - b_{\sigma_{\psi,2}(i)})^2 \right) \right] \\
&= \mathbb{E}_{\psi \sim \mathcal{U}(\mathbb{S}^{d+k-1})} \left[ \exp\left( -\gamma \sum_i^N \left[ \left( \underbrace{\psi_1^\top z_x^{\sigma_1(i)} - \psi_1^\top z_y^{\sigma_2(i)}}_{K_{\psi,1}} + \underbrace{\psi_2^\top pos(\sigma_1(i)) - \psi_2^\top pos(\sigma_2(i))}_{K_{\psi,2}} \right)^2 \right] \right) \right] \\
&= \mathbb{E}_{\psi \sim \mathcal{U}(\mathbb{S}^{d+k-1})} \left[ \exp\left( -\gamma \sum_i^N \left[ K_{\psi,1}^2 + 2 K_{\psi,1} K_{\psi,2} + K_{\psi,2}^2 \right] \right) \right].
\end{aligned}
$$
$$(12)$$

The $K_{\psi,1}^2$ term and the $K_{\psi,2}^2$ term in Equation (12) are the distance regarding feature space and the temporal distance in terms of position with respect to the projecting direction $\psi$. The temporal-similarity score is jointly optimized with the likelihood objective function in Equation (1) to train the audio captioning model

$$
\mathcal{L} = \mathcal{L}_{MLE}(x, y) + \mathcal{UK}_\gamma(Z_x, Z_y; 2).
$$
$$(13)$$

**Inference stage.** As extensively discussed in the literature, likelihood decoding is suffering from exposure bias (An et al., 2022; Su et al., 2022). A solution is to utilize stochastic decoding, such as top-k or nucleus sampling (Holtzman et al., 2020)methods, to mitigate the harmful effect of exposure bias (Arora et al., 2022b). We propose to leverage the temporal-similarity score based on the USW-RBF between the latent representation of audio and generated captions as a decoding criterion. As demonstrated in the Figure 1, the pretrained audio captioning model generates $\mathcal{B}$ candidate captions by stochastic decoding methods, and the most likely caption is chosen as follows

$$
\mathbf{y}^* = \underset{\mathbf{y_i} \in \mathcal{B}}{\arg\max} \{ (1-\alpha) p(\mathbf{y_i}|x) + \alpha . \mathcal{UK}_\gamma(Z_x, Z_y; 2)
$$
$$(14)$$

where $Z_x, Z_{\mathbf{y}_i}$ denote the latent representation of audio and generated captions outputted from the encoder and decoder models, respectively. The coefficient $0 < \alpha < 1$ is set to $0.5$ in the most case. The first term of the decoding objective is the likelihood score of a generated caption, which measures the confidence of the audio captioning model. The second term measures the similarity in terms of the latent representation of audio and generated captions.

# 4 RELATED WORK

**Audio captioning.** The audio captioning task can be formulated as a conditional text generation task, therefore, the prior works utilize the maximum likelihood estimation method to train audio captioning models (Mei et al., 2021; 2024; Sun et al., 2023; Kim et al., 2022; Deshmukh et al., 2023). There are two popular architectures for audio captioning models: encoder-decoder architecture Mei et al. (2024); Kim et al. (2024) and prefix-tuning architecture (Deshmukh et al., 2023; Kim et al., 2023). Although both architectures are effective in generating plausible captions, they suffer from the inherent weakness of the MLE training method: exposure bias. Some recent works deal with exposure bias by leveraging a regularization (Zhang et al., 2023; Deshmukh et al., 2024), contrastive loss. The contrastive regularization can slightly remedy the exposure bias issue for audio captioning models. Another technique to combat with exposure bias is to utilize stochastic decoding methods (Arora et al., 2022a). (Su et al., 2022) proposed a contrastive search framework with

stochastic decoding methods to alleviate text degeneration for conditional text generation. The contrastive search framework is yet successful to deal with exposure bias for text generation, it can not be directly applied for audio captioning task. The reason is that the contrastive score is not able to take temporal information of acoustic and linguistic features into account. To deal with the shortcomings of the contrastive framework, we develop a new framework, called ACUS, which can handle the temporal information between acoustics and linguistic modalities when measuring the similarity score and alleviate exposure bias at the inference stage for audio captioning.

**Wasserstein distance.** Wasserstein distance is a metric to measure the discrepancy between two distributions. There are enormous applications of the Wasserstein distance for multimodal learning, such as audio-text retrieval (Luong et al., 2024), multimodal representation learning (Tsai et al., 2019), and multimodal alginment (Lee et al., 2019). The prior work (Su & Hua, 2017) proposed an order-preserving Wasserstein distance between sequences by incorporating a soft-monotonic alignment prior for optimal matching, however, it still suffers from dimensionality curse and a strict monotonic alignment across modalities. Although the Wasserstein distance is capable of measuring the cross-modality distance, it suffers from the dimensionality curse. In this work, we develop the USW-RBF kernel equipped with positional encoding to deal with the dimensionality curse and the strict monotonic alignment issue of measuring cross-modal similarity for audio captioning.

## 5 EXPERIMENTS

We design experiments to demonstrate the effectiveness of our proposed method in mitigating exposure bias in the audio captioning task. We conduct quantitative experiments on two datasets: Audiocaps (Kim et al., 2019) and Clotho (Drossos et al., 2020) to answer the question of whether our proposed method is capable of alleviating exposure bias in the audio captioning task. We further conduct qualitative experiments on audio-text retrieval tasks and subjective evaluation to show the high-quality of generated captions. Finally, we perform ablation studies on the choice of similarity metric and positional embedding techniques. The ablation studies show that the proposed metric outperforms both Wasserstein distance, DTW, and soft-DTW in measuring the similarity between latent representation of audio and generated captions. These studies also show that rotary positional embedding is the most well-suited positional embedding technique for incorporating temporal information for audio-captioning. Baselines and implementation details can be found in Appendix A.2.

**Evaluation metrics.** We evaluate baselines and two backbone models, Enclap and ACT, for our proposed framework by widely used evaluation metrics for audio captioning, including METEOR (Banerjee & Lavie, 2005), ROUGE-L (Lin, 2004), CIDEr (Vedantam et al., 2014), SPICE (Anderson et al., 2016), and SPIDEr (Liu et al., 2016). In addition, we evaluate the quality of generated audio captions by performing a text-to-audio retrieval task leveraging the pretrained CLAP (Wu et al., 2023) model. If a generated caption and a given audio are highly similar to each other, the CLAP model is able to retrieve the audio by using the generated caption. We further measure the lexical diversity and caption length in generated captions to measure the degeneration of captions. We also conduct a subjective evaluation to evaluate the quality of generated captions in terms of discretiveness, correctness, and fluency.

### 5.1 QUANTITATIVE EXPERIMENTS

To assess the performance of our proposed method for audio captioning, we performed quantitative experiments on Audiocaps and Clotho. The experimental results are shown in the Table. 1. All baseline models utilize deterministic decoding methods, the beam search decoding, therefore their performance is not variant in each evaluation. On the other hand, the contrastive method and our framework utilize stochastic decoding methods, such as the nucleus and top-k samplings, thus its performance varies for each evaluation. To make a fair comparison, we evaluate both our framework and contrastive method 5 times and report the average performance and standard deviation. It is clear to see that our proposed method outperforms all baseline models in terms of automated metrics on the AudioCaps test set. Specifically, our proposed framework significantly improves the quality of generated captions for the Enclap backbone model. There is a significant improvement regarding the statistical metrics SPICE, METEOR, CIDEr, and ROUGE-L. These results prove that our proposed method is able to mitigate the exposure bias for audio captioning models during inference. Furthermore, there is a significant performance gain regarding the SPICE score, from 0.186 to 0.192. Since

Table 1: The quantitative evaluation of proposed method with baselines using objective metrics on AudioCaps and Clotho datasets. The ACUS and contrastive frameworks utilize stochastic decoding methods during the inference stage, therefore, we report the average performance and standard deviation for these methods.

| Dataset | Method | METEOR | ROUGE_L | CIDEr | SPICE | SPIDEr |
|---|---|---|---|---|---|---|
| AudioCaps | ACT | 0.222 | 0.468 | 0.679 | 0.160 | 0.420 |
| | LHDFF | 0.232 | 0.483 | 0.680 | 0.171 | 0.426 |
| | CNN14-GPT2 | 0.240 | 0.503 | 0.733 | 0.177 | 0.455 |
| | BART-tags | 0.241 | 0.493 | 0.753 | 0.176 | 0.465 |
| | Pengi | 0.232 | 0.482 | 0.752 | 0.182 | 0.467 |
| | AL-MixGen | 0.242 | 0.502 | 0.769 | 0.181 | 0.475 |
| | WavCaps | 0.250 | - | 0.787 | 0.182 | 0.485 |
| | Enclap | 0.254 | 0.5 | 0.77 | 0.186 | 0.48 |
| | Enclap + CL | $0.257 \pm 0.001$ | $0.496 \pm 0.001$ | $0.768 \pm 0.003$ | $0.19 \pm 0.001$ | $0.481 \pm 0.003$ |
| | Our method | $\mathbf{0.262 \pm 0.001}$ | $\mathbf{0.509 \pm 0.001}$ | $\mathbf{0.807 \pm 0.003}$ | $\mathbf{0.192 \pm 0.001}$ | $\mathbf{0.5 \pm 0.002}$ |
| Clotho | CLIP-AAC | 0.168 | 0.372 | 0.394 | 0.115 | 0.254 |
| | LHDFF | 0.175 | 0.378 | 0.408 | 0.122 | 0.265 |
| | MAAC | 0.174 | 0.377 | 0.419 | 0.119 | 0.269 |
| | Enclap | 0.182 | 0.38 | 0.417 | 0.13 | 0.273 |
| | Enclap + CL | $0.185 \pm 0.001$ | $0.376 \pm 0.002$ | $0.405 \pm 0.001$ | $0.131 \pm 0.002$ | $0.271 \pm 0.002$ |
| | Our method | $\mathbf{0.186 \pm 0.001}$ | $\mathbf{0.38 \pm 0.001}$ | $\mathbf{0.419 \pm 0.004}$ | $\mathbf{0.133 \pm 0.001}$ | $\mathbf{0.275 \pm 0.003}$ |

Table 2: Experiments of our framework on the AudioCaps dataset with two encoder-decoder audio captioning models, ACT and Enclap, to show the effectiveness of the ACUS framework.

| Model | Decoding | METEOR | ROUGE_L | CIDEr | SPICE | SPIDEr |
|---|---|---|---|---|---|---|
| ACT | Beam(k=5) | 0.222 | 0.468 | 0.679 | 0.160 | 0.420 |
| | Top-p(p=0.5) | $\mathbf{0.245 \pm 0.001}$ | $\mathbf{0.49 \pm 0.002}$ | $\mathbf{0.714 \pm 0.01}$ | $\mathbf{0.180 \pm 0.002}$ | $\mathbf{0.446 \pm 0.005}$ |
| | Top-k(k=5) | $0.241 \pm 0.001$ | $0.482 \pm 0.001$ | $0.687 \pm 0.002$ | $0.178 \pm 0.001$ | $0.432 \pm 0.002$ |
| | Temp(temp=1.0) | $0.235 \pm 0.002$ | $0.478 \pm 0.002$ | $0.677 \pm 0.004$ | $0.175 \pm 0.002$ | $0.426 \pm 0.002$ |
| Enclap | Beam(k=5) | 0.254 | 0.5 | 0.77 | 0.186 | 0.48 |
| | Top-p(p=0.7) | $0.262 \pm 0.002$ | $\mathbf{0.509 \pm 0.001}$ | $\mathbf{0.807 \pm 0.004}$ | $0.192 \pm 0.001$ | $\mathbf{0.501 \pm 0.002}$ |
| | Top-k(k=5) | $0.262 \pm 0.004$ | $0.508 \pm 0.003$ | $0.801 \pm 0.01$ | $\mathbf{0.193 \pm 0.001}$ | $0.497 \pm 0.005$ |
| | Temp(temp=1.0) | $\mathbf{0.265 \pm 0.002}$ | $0.483 \pm 0.002$ | $0.718 \pm 0.011$ | $0.191 \pm 0.002$ | $0.49 \pm 0.003$ |

the SPICE score captures the semantic similarity between generated and ground-truth captions, the proposed method is able to generate better semantically similar captions with reference. A similar improvement regarding objective metrics is observed for the Clotho dataset. The improvement is insignificant due to the diversity of reference captions in the Clotho dataset for automated metrics like ROUGE_L and CIDEr that rely on measuring statistical overlap between predicted and reference captions.

In Table 2, we conducted the experiment on the diverse audio captioning backbones, the Enclap and ACT models, for the proposed method. The Enclap model is a encoder-decoder model which consists of a pretrained audio encoder from the CLAP model (Wu et al., 2023) and a pretrained BART decoder model. The ACT model is also a encoder-decoder model, which includes a vision transformer encoder pretrained on the AudioSet dataset and a transformer decoder model. The performance of backbone models with beam search decoding is substantially enhanced by our proposed approach when decoded with stochastic decoding techniques. The nucleus sampling technique with our method achieves the highest performance gain for both backbone models, while the stochastic decoding with temperature shows a little improvement. Especially, there is a slight drop in the CIDEr metric using stochastic decoding with temperature. The experimental results show the importance of controlling stochasticness when decoding to mitigate exposure bias. We also carry out ablation studies for choosing hyperparameters for stochastic decoding methods using our framework, and the results are reported in the Appendix A.3.

## 5.2 QUALITATIVE EXPERIMENTS

We carry out qualitative experiments to examine the capability of alleviating exposure bias and caption degeneration of our proposed method. The pretrained CLAP (Wu et al., 2023) model is used for the text-to-audio self-retrieval experiments. As shown in Table 3, our method is able to enhance the caption length and lexical diversity of generated captions on both datasets compared to the contrastive learning method. Caption length and lexical diversity increase from 7.63 to 8.14

Table 3: Qualitative experiments of baseline methods and our proposed method on AudioCaps and Clotho datasets. For human captions, we evaluate five ground-truth captions and report mean and standard deviation results.

| Dataset | Method | Caption Length | Lexical Diversity | Text-to-audio retrieval | | |
|---------|--------|----------------|-------------------|------|------|------|
| | | | | R@1 | R@5 | R@10 |
| AudioCaps | Enclap | 7.52 | 7.06 | 29.2 | 70 | 85 |
| | Enclap + CL | $7.63 \pm 0.01$ | $7.21 \pm 0.015$ | $30.4 \pm 0.13$ | $71.3 \pm 0.27$ | $86.2 \pm 0.32$ |
| | Enclap + ACUS | $\mathbf{8.66 \pm 0.012}$ | $\mathbf{7.96 \pm 0.021}$ | $\mathbf{32.2 \pm 0.21}$ | $\mathbf{73.6 \pm 0.42}$ | $\mathbf{88.36 \pm 0.5}$ |
| | Human | $10.3 \pm 0.128$ | $9.48 \pm 0.124$ | $35.9 \pm 1.69$ | $74 \pm 1.2$ | $85.9 \pm 1.27$ |
| Clotho | Enclap | 11.23 | 10.13 | 9.3 | 30.4 | 43.1 |
| | Enclap + CL | $11.45 \pm 0.027$ | $10.24 \pm 0.024$ | $9.7 \pm 0.28$ | $31.2 \pm 0.35$ | $47.6 \pm 0.49$ |
| | Enclap + ACUS | $\mathbf{12.14 \pm 0.032}$ | $\mathbf{10.83 \pm 0.027}$ | $\mathbf{11.3 \pm 0.34}$ | $\mathbf{33.54 \pm 0.55}$ | $\mathbf{48.7 \pm 0.66}$ |
| | Human | $11.31 \pm 0.11$ | $10.57 \pm 0.06$ | $15.5 \pm 0.91$ | $39.7 \pm 1.25$ | $52.6 \pm 2.22$ |

Table 4: Human evaluation results on two subsets of 50 audio of AudioCaps and Clotho test set. Each method generates a single caption given an audio, while one human caption is randomly selected from five ground-truth captions. ∗ are statistically significant results with Sign-test ($p < 0.05$).

| Method | AudioCaps | | | Clotho | | |
|--------|-----------|---|---|--------|---|---|
| | Descriptiveness | Correctness | Fluency | Descriptiveness | Correctness | Fluency |
| Enclap + MLE | 4.02 | 4.24 | 4.95 | 3.56 | 3.34 | 4.66 |
| Enclap + CL | 4.06 | 4.47 | 4.97 | 3.62 | 3.45 | 4.85 |
| Enclap + ACUS | $\mathbf{4.28^{*}}$ | $\mathbf{4.54^{*}}$ | $\mathbf{4.98}$ | $\mathbf{3.7^{*}}$ | $\mathbf{3.6^{*}}$ | $\mathbf{4.92}$ |
| Human caption | 4.56 | 4.76 | 4.88 | 3.96 | 3.94 | 4.66 |
| Agreement (Fleiss kappa $\kappa$) | 0.47 | 0.52 | 0.65 | 0.42 | 0.46 | 0.58 |

and from 7.21 to 7.52 on AudioCaps dataset, respectively. Furthermore, the caption to audio self-retrieval experiments show that our proposed method is able to generate high-quality captions which are beneficial to retrieving corresponding audio. These results show that the proposed framework can mitigate the exposure bias for audio captioning tasks and generate high-quality captions.

**Human evaluation.** We conduct a human evaluation to better assess the quality of generated captions. We randomly choose 50 audio from AudioCaps and Clotho test data. Captions are generated for each audio by using different methods: maximum likelihood estimation (MLE), contrastive framework, and the ACUS framework. The MLE method utilizes a deterministic decoding method, beam search with a beam size of 5, while contrastive learning and the proposed method utilize a stochastic decoding method, top-p sampling with $p = 0.7$ to generate 30 candidate captions. The most suitable caption is chosen based on Equation (5) for contrastive learning and Equation (14) for the proposed method. We recruit five annotators, who are asked to independently assess the quality of a given caption following a 5-point Likert scale for three aspects

- **Descriptiveness:** Whether the caption is descriptive enough, describe all audio events in the given audio and their temporal relationships.
- **Correctness:** Whether the caption is correct, all audio events occur in the given audio.
- **Fluency:** Whether the caption is fluent and easy to understand as human written.

Table 4 shows the human valuation results on three aspects for Audiocaps and Clotho datasets. The inter-annotator agreement is shown in the last row measured by the Fleiss Kappa score (Fleiss, 1971). On both datasets, our method is capable of generating more descriptive and correct captions compared to baseline models trained with MLE and contrastive learning objectives. Also, all generated captions are more fluent than human-written captions. The rationale behind it is that humans focus more on audio content rather than fluency. On the other hand, audio captioning models leverage pretrained language models as the decoder, therefore, they can generate coherence captions but less focus on describing audio content. The qualitative examples can be found in Appendix A.4.

## 5.3 ABLATION STUDIES

Table 5 shows the ablation study on choosing similarity metrics for measuring audio and caption similarity. The DTW and soft- DTW are ineffective in measuring the similarity across acoustic and linguistic modality. Therefore, there is a decrease in performance compared with the baseline

Table 5: Ablation study on the effectiveness of the similarity score based on the USW-RBF kernel for audio captioning on the AudioCaps dataset with the Enclap backbone. All similarity metrics are evaluated using our proposed framework with top-p sampling with $p = 0.7$.

| Similarity score | METEOR | ROUGE_L | CIDEr | SPICE | SPIDEr |
|---|---|---|---|---|---|
| w/o score + beam search | 0.254 | 0.5 | 0.77 | 0.186 | 0.48 |
| DTW | $0.248 \pm 0.001$ | $0.492 \pm 0.001$ | $0.762 \pm 0.002$ | $0.184 \pm 0.001$ | $0.473 \pm 0.003$ |
| soft-DTW | $0.251 \pm 0.002$ | $0.497 \pm 0.002$ | $0.764 \pm 0.004$ | $0.187 \pm 0.001$ | $0.475 \pm 0.003$ |
| Wasserstein w/ PE | $0.262 \pm 0.001$ | $0.499 \pm 0.007$ | $0.756 \pm 0.005$ | $\mathbf{0.194 \pm 0.001}$ | $0.475 \pm 0.003$ |
| Our score | $\mathbf{0.262 \pm 0.001}$ | $\mathbf{0.509 \pm 0.001}$ | $\mathbf{0.807 \pm 0.003}$ | $0.193 \pm 0.001$ | $\mathbf{0.5 \pm 0.002}$ |

Table 6: Ablation study on the effectiveness of positional embedding techniques on the AudioCaps dataset with the Enclap backbone for our proposed framework. The decoding method is top-p sampling with $p = 0.7$.

| PE method | METEOR | ROUGE_L | CIDEr | SPICE | SPIDEr |
|---|---|---|---|---|---|
| w/o PE | $0.259 \pm 0.002$ | $0.501 \pm 0.003$ | $0.787 \pm 0.005$ | $0.191 \pm 0.002$ | $0.485 \pm 0.003$ |
| Absolute PE | $0.26 \pm 0.002$ | $0.502 \pm 0.001$ | $0.789 \pm 0.002$ | $0.192 \pm 0.001$ | $0.490 \pm 0.002$ |
| Rotary PE | $\mathbf{0.262 \pm 0.001}$ | $\mathbf{0.509 \pm 0.001}$ | $\mathbf{0.807 \pm 0.003}$ | $\mathbf{0.193 \pm 0.001}$ | $\mathbf{0.5 \pm 0.002}$ |

method with beam search decoding. The hypothesis is that the constraint for monotonic alignment between acoustic and linguistic embedding is too strict for measuring the distance between two modalities. Our score and the Wasserstein distance relax the monotonic alignment constraint when computing cross-modality similarity. Both our score and the Wasserstein distance are equipped with the positional embedding to consider temporal information when measuring similarity across modalities. Relaxing the monotonic alignment and incorporating positional embedding(PE) shows a significant performance gain regarding METEOR and SPICE metrics with the Wasserstein distance, $0.254$ to $0.262$ and $0.186$ to $0.194$, respectively. Although the Wasserstein distance with positional embedding is effective in measuring acoustic and linguistic similarity, it possesses a weakness: the dimensionality curse. Thus, there is still a gap in calculating similarity across acoustic and linguistic modalities. As mentioned in (Nguyen & Ho, 2022; Nietert et al., 2022; Nadjahi et al., 2020), the sliced Wasserstein does not suffer from the dimensionality curse. The performance of the USW-RBF score acquires a performance gain with all evaluation metrics, which reflects that the sliced Wasserstein with positional embedding is the most effective score for computing audio and caption similarity. The ablation study on the number of Monte Carlo samples $L$ for estimating the USW-RBF is shown in Table 8 in Appendix A.3.

We conducted an ablation study on the effectiveness of positional embedding techniques for our method. As shown in Table 6, the rotary positional embedding technique outperforms the absolute positional embedding technique regarding all evaluation metrics. The rotary positional embedding (PE) technique outperforms both without PE and the absolute PE technique regarding all objective metrics. These empirical results indicate that the rotary PE technique is the most suitable method for the ACUS framework to account for temporal information when measuring cross-modal similarity.

## 6 CONCLUSION

We introduce the ACUS framework for alleviating text degeneration for the audio captioning task. Furthermore, we develop the USW-RBF kernel equipped with the rotary positional embedding. The USW-RBF is an unbias kernel, thus, it is compatible with stochastic gradient optimization algorithms, and its approximation error decreases at a parametric rate of $\mathcal{O}(L^{-1/2})$. Our experiments demonstrate that our framework is able to mitigate the text degeneration issue for audio captioning models and outperforms baseline methods in terms of quantitative and qualitative evaluations. We further find that the nucleus sampling technique is the best decoding method to generate descriptive and correct captions from pretrained audio captioning models.

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

# A APPENDIX

## A.1 PROOFS

### A.1.1 PROOF OF PROPOSITION 1

From Theorem 4 in (Kolouri et al., 2016), we have $\mathcal{K}_\gamma(\mu, \nu) = \exp\left(\gamma W_2^2(\mu, \nu)\right)$ is a positive definite kernel for $\mu$ and $\nu$ are two absolute continuous distribution in one-dimension. It means that for all $n > 1$ one-dimensional absolute continuous distributions $\mu_1, \ldots, \mu_n$ and $c_1, \ldots, c_n \in \mathbb{R}$, we have:

$$\sum_{i=1}^{n} \sum_{j=1}^{n} c_i c_j \exp(\gamma W_2^2(\mu_i, \mu_j)) > 0.$$

When $\mu$ and $\nu$ are absolute continuous distributions in $d > 1$ dimension, given $\psi \in \mathbb{S}^{d-1}$, $\psi\sharp\mu$ and $\psi\sharp\nu$ are also absolute continuous distribution since the pushfoward function $f_\psi(x) = \psi^\top x$ is a absolute continuous function. As a result, or all $n > 1$ one-dimensional absolute continuous distributions $\mu_1, \ldots, \mu_n$ and $c_1, \ldots, c_n \in \mathbb{R}$, we have:

$$\sum_{i=1}^{n} \sum_{j=1}^{n} c_i c_j \exp(\gamma W_2^2(\psi\sharp\mu_i, \psi\sharp\mu_j)) > 0.$$

Taking the expectation with respect to $\psi \sim \mathcal{U}(\mathbb{S}^{d-1})$, we have:

$$\mathbb{E}\left[\sum_{i=1}^{n} \sum_{j=1}^{n} c_i c_j \exp(\gamma W_2^2(\psi\sharp\mu_i, \psi\sharp\mu_j))\right] > 0.$$

It is equivalent to

$$\sum_{i=1}^{n} \sum_{j=1}^{n} c_i c_j \mathbb{E}\left[\exp(\gamma W_2^2(\psi\sharp\mu_i, \psi\sharp\mu_j))\right] > 0,$$

which yields the desired inequality:

$$\sum_{i=1}^{n} \sum_{j=1}^{n} c_i c_j \mathcal{U}\mathcal{K}_\gamma(\mu_i, \mu_j; 2) > 0.$$

Therefore, the USW-RBF kernel is positive definite for $p = 2$.

### A.1.2 PROOF OF PROPOSITION 2

We first recall the definition of SW-RBF (Equation (8)) and the definition of USW-RBF (Definition 1.

$$\mathcal{K}_\gamma(\mu, \nu) = \exp\left(-\gamma SW_p^p(\mu, \nu)\right),$$
$$\mathcal{U}\mathcal{K}_\gamma(\mu, \nu; p) = \mathbb{E}_{\psi\sim\mathcal{U}(\mathbb{S})^{d-1}}\left[\exp\left(-\gamma W_p^p(\psi\sharp\mu, \psi\sharp\nu)\right)\right].$$

Applying Jensen's inequality, we have:

$$\begin{aligned}
\mathcal{U}\mathcal{K}_\gamma(\mu, \nu; p) &= \mathbb{E}_{\psi\sim\mathcal{U}(\mathbb{S})^{d-1}}\left[\exp\left(-\gamma W_p^p(\psi\sharp\mu, \psi\sharp\nu)\right)\right] \\
&\geq \exp\left(\mathbb{E}_{\psi\sim\mathcal{U}(\mathbb{S})^{d-1}}\left[-\gamma W_p^p(\psi\sharp\mu, \psi\sharp\nu)\right]\right) \\
&= \exp\left(\gamma\mathbb{E}_{\psi\sim\mathcal{U}(\mathbb{S})^{d-1}}\left[-W_p^p(\psi\sharp\mu, \psi\sharp\nu)\right]\right) \\
&= \exp\left(-\gamma SW_p^p(\mu, \nu)\right) = \mathcal{K}_\gamma(\mu, \nu),
\end{aligned}$$

which completes the proof.

### A.1.3 PROOF OF PROPOSITION 3

(i) For the unbiasedness, we check:

$$\mathbb{E}[\widehat{UK}_\gamma(\mu, \nu; p, L)] = \mathbb{E}\left[\frac{1}{L}\sum_{l=1}^{L} \exp\left(-\gamma W_p^p(\psi_l \sharp \mu, \psi_l \sharp \nu)\right)\right]$$

$$= \frac{1}{L}\sum_{l=1}^{L} \mathbb{E}\left[\exp\left(-\gamma W_p^p(\psi_l \sharp \mu, \psi_l \sharp \nu)\right)\right]$$

$$= \frac{1}{L}\sum_{l=1}^{L} \mathcal{UK}_\gamma(\mu, \nu; p) = \mathcal{UK}_\gamma(\mu, \nu; p),$$

where the last equality is due to the fact that $\psi_1, \ldots, \psi_L \overset{i.i.d}{\sim} \mathcal{U}(\mathbb{S}^{d-1})$.

(ii) Using the Holder's inequality, we have, we have:

$$\mathbb{E}\left[\left|\widehat{UK}_\gamma(\mu, \nu; p, L) - \mathcal{UK}_\gamma(\mu, \nu; p)\right|\right]$$

$$\leq \sqrt{\mathbb{E}\left[\left|\widehat{UK}_\gamma(\mu, \nu; p, L) - \mathcal{UK}_\gamma(\mu, \nu; p)\right|^2\right]}.$$

From (i), we have $\mathbb{E}[\widehat{UK}_\gamma(\mu, \nu; p, L)] = \mathcal{UK}_\gamma(\mu, \nu; p)$, hence,

$$\mathbb{E}\left[\left|\widehat{UK}_\gamma(\mu, \nu; p, L) - \mathcal{UK}_\gamma(\mu, \nu; p)\right|\right] \leq \sqrt{\mathrm{Var}\left[\widehat{UK}_\gamma(\mu, \nu; p, L)\right]}$$

$$= \sqrt{\mathrm{Var}\left[\frac{1}{L}\sum_{l=1}^{L} \exp\left(-\gamma W_p^p(\psi_l \sharp \mu, \psi_l \sharp \nu)\right)\right]}$$

$$= \sqrt{\frac{1}{L^2}\sum_{l=1}^{L} \mathrm{Var}\left[\exp\left(-\gamma W_p^p(\psi_l \sharp \mu, \psi_l \sharp \nu)\right)\right]}$$

$$= \sqrt{\frac{1}{L}\mathrm{Var}\left[\exp\left(-\gamma W_p^p(\psi \sharp \mu, \psi \sharp \nu)\right)\right]},$$

which completes the proof.

### A.2 IMPLEMENTATION DETAILS

**Baselines.** We compare against all state-of-the-art audio captioning models on Audiocaps and Clotho datasets. The ACT (Mei et al., 2021) audio captioning model leverages a vision transformer encoder pretrained on the AudioSet (Gemmeke et al., 2017) dataset for sound-event classification. LHDFF (Sun et al., 2023) utilizes residual the PANNs encoder to fuse low and high dimensional features in Mel-spectrogram. CNN14-GPT2 (Kim et al., 2023) and Pengi (Deshmukh et al., 2023) apply prefix-tuning method for the pretrained GPT2 (Radford et al., 2019). The BART-tags (Gontier et al., 2021) model generates audio captions relying on predefined audio tags from the AudioSet dataset. AL-MixGen (Kim et al., 2022) leverages the ACT backbone trained using audio-language mixup augmentation and test-time augmentation at the inference phase. Wavcaps Mei et al. (2024) is the HTSAT-BART model Chen et al. (2022b) fine-tuned on numerous weakly-labeled data which is generated by using large language models. We choose a subset of models evaluated on the Clotho dataset without complex training methods, such as ensemble training, to ensure a fair comparison. The CLIP-AAC (Chen et al., 2022a), MAAC (Ye et al., 2021), P-LocalAFT(Xiao et al., 2022), and Graph-AC (Xiao et al., 2023) are the baselines evaluated on Clotho dataset.

**Enclap backbone.** We follow the original settings in (Kim et al., 2024) to train the large Enclap backbone for AudioCaps and Clotho dataset. The training objective is described in Eq. 13, in which the MLE and temporal-similarity are jointly optimized to train the Enclap model. The training coefficient $\alpha$ is set to 0.1 for both two datasets. The Adam optimizer with $\beta_1 = 0.9$, $\beta_2 = 0.999$,

and a weight decay coefficient of $0.01$ is used to train the model for both datasets. For AudioCaps, we use a batch size of 64 and warm up for 2000 steps before reaching the peak learning rate at $lr = 2e^{-5}$. For Clotho, we use a batch size of 48 with the gradient accumulation step of 2 and warm up for 1000 steps before reaching the peak learning rate at $lr = 2e^{-5}$. We perform a grid search for the hyperparameter $\gamma = \{0.5, 1.5, 2.5, 3.5\}$ for the temporal-similarity metric. We choose the best value of $\gamma$, which is 2.5 and 1.5 for the AudioCaps and Clotho datasets, respectively. We also perform a grid search for the stochastic decoding methods at the inference state to choose the best decoding hyperparameters for each stochastic decoding method, $p = \{0.5, 0.6, 0.7, 0.8, 0.9\}$ for top-p sampling, $k = \{3, 4, 5\}$ for top-k sampling, and $temp = \{1.1, 1.2, 1.3, 1.4, 1.5\}$ for temperature sampling. The best results with optimal decoding hyperparameters are reported in Table 2.

**ACT backbone.** We follow the original settings in (Mei et al., 2021) to train the audio captioning transformer (ACT) backbone on the AudioCaps dataset. We use a batch size of 32 and warm up for five epochs before reaching the peak learning rate at $lr = 1e^{-4}$. We use the training objective function in Equation (13) with training coefficient $\alpha = 0.1$ and the bandwidth for the temporal-similarity metric $\gamma = 2.5$. We also perform a grid search for stochastic decoding methods at the inference state to choose the best hyperparameters for each stochastic decoding method, $p = \{0.5, 0.6, 0.7, 0.8, 0.9\}$ for top-p sampling, $k = \{3, 4, 5\}$ for top-k sampling, and $temp = \{1.1, 1.2, 1.3, 1.4, 1.5\}$ for temperature sampling. The best results with optimal decoding hyperparameters are reported in Table 2.

**DTW and soft-DTW as dissimilarity metric.** DTW is a non-parametric distance which measures an optimal monotonic alignment between two time series of different lengths. The definition of DTW is defined as follows

$$DTW(C(Z_X, Z_Y)) = \min_{A \in \mathcal{A}(m,n)} \langle A, C \rangle, \tag{15}$$

where $Z_X \in \mathbb{R}^{n \times d}$ and $Z_y \in \mathbb{R}^{m \times d}$ are two $d-$dimensional sequences of audio and text hidden representation. The cost matric between them is denoted as $C(Z_X, Z_Y)$, in which its element is computed as $c_{i,j} = \frac{1}{2}||z_x^i - z_y^j||_2^2$. We denote $\mathcal{A}(m,n) \subset {0,1}^{m \times n}$ as a set of all such monotonic alignment matrices. The soft- DTW is a variant of DTW which is compute as follow

$$SDTW_\gamma(C(X,Y)) = -\gamma \log \sum_{A \in \mathcal{A}(m,n)} \exp(-\langle A, C \rangle / \gamma), \tag{16}$$

where $\gamma$ is a parameter which controls the tradeoff between approximation and smoothness.

**Wasserstein distance as dissimilarity metric.** The Wasserstein distance measures the similarity between two probabilities over a metric space. We denote the distribution $\mu = \frac{1}{N} \sum_{i=1}^{N} \delta_{z_x^i}$ and $\nu = \frac{1}{M} \sum_{j=1}^{M} \delta_{z_y^j}$ as the empirical distribution of hidden representation of audio and caption, respectively. The Wasserstein between audio and text hidden representation is defined as

$$W(\mu, \nu) = \min_{\pi \in \Pi(\mu,\nu)} \sum_{i=1}^{N} \sum_{j=1}^{M} \pi_{i,j}||z_x^i - z_y^j||^2, \tag{17}$$

where $\Pi(\mu, \nu) = \{\pi \in \mathbb{R}^{n \times m} | \pi 1_m = 1_n/n, \pi^T 1_m/m\}$ denotes all set of feasible coupling between $\mu$ and $\nu$.

## A.3 ABLATION STUDIES

The ablation study for the bandwidth parameter $\gamma$ is shown in the Table 7. To simplify the hyperparameter tuning, we perform beam search decoding to evaluate the performance of different values of the bandwidth parameter on two datasets. The optimal values for the bandwidth parameter are $\gamma = 2.5$ and $\gamma = 1.5$ on Audiocaps and Clotho datasets, respectively. Furthermore, ablation studies on choosing hyperparameters for stochastic decoding methods on Audiocaps dataset are demonstrated in the Figure 2. The SPIDEr metric is chosen as the criterion for hyperparameter selection for stochastic decoding methods, like nucleus, top-k, and temperature samplings. According to the experiments, nucleus sampling acquires the highest performance regarding the SPIDEr metric with $p = 0.7$. Therefore, we choose nucleus sampling with $p = 0.7$ to conduct experiments for our proposed framework.

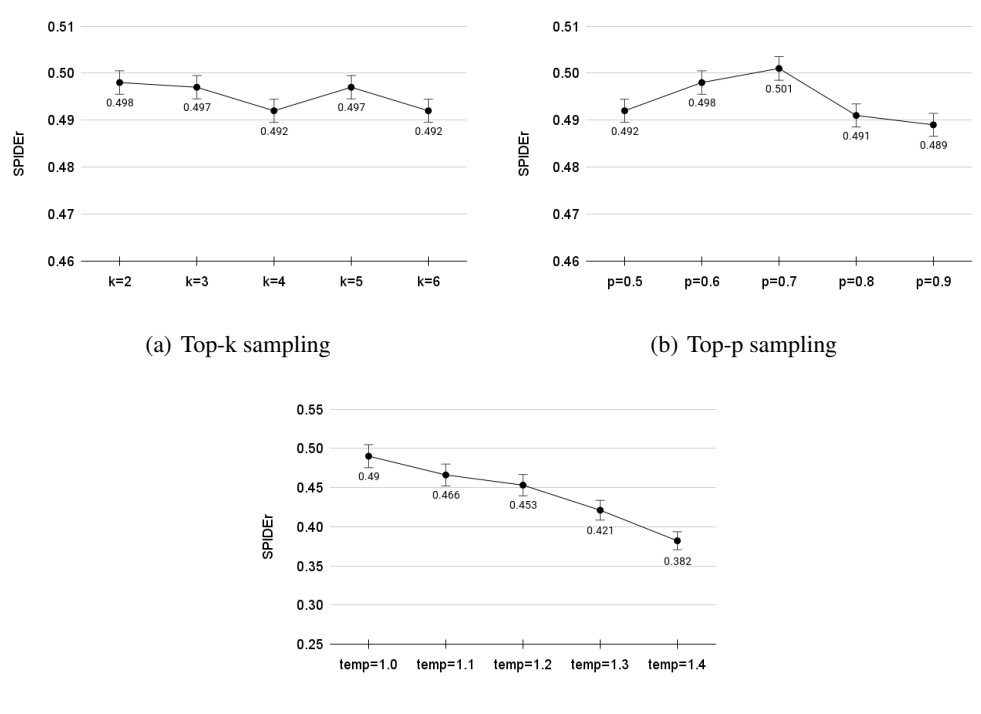

(a) Top-k sampling

(b) Top-p sampling

(c) Temperature sampling

Figure 2: Ablatio studies for sampling hyperparmeters of stochastic sampling methods of the Enclap backbone on the AudioCaps dataset. The SPIDEr metric is chosen for sampling hyperparameters tuning since it is the combination of the SPICE and CIDEr evaluation metrics

Table 7: Ablation study for the bandwidth hyperparameter selection on AudioCaps and Clotho datasets. To simplify the hyperparameter selection, we conduct experiments with beam search decoding for choosing the best bandwidth parameter $\gamma$ for each dataset.

| Dataset | $\gamma$ | METEOR | ROUGE_L | CIDEr | SPICE | SPIDEr |
|---|---|---|---|---|---|---|
| AudioCaps | $\gamma = 0.5$ | 0.251 | 0.493 | 0.755 | 0.186 | 0.470 |
| | $\gamma = 1.0$ | 0.254 | 0.495 | 0.773 | 0.185 | 0.479 |
| | $\gamma = 1.5$ | 0.254 | 0.497 | 0.771 | 0.187 | 0.479 |
| | $\gamma = 2.0$ | 0.251 | 0.495 | 0.756 | 0.183 | 0.469 |
| | $\gamma = 2.5$ | 0.253 | **0.502** | **0.79** | **0.188** | **0.492** |
| | $\gamma = 3.0$ | **0.254** | 0.50 | 0.787 | 0.185 | 0.487 |
| Clotho | $\gamma = 0.5$ | 0.186 | 0.380 | 0.433 | 0.134 | 0.283 |
| | $\gamma = 1.0$ | 0.185 | 0.381 | 0.431 | 0.134 | **0.284** |
| | $\gamma = 1.5$ | **0.186** | **0.382** | **0.433** | **0.137** | 0.283 |
| | $\gamma = 2.0$ | 0.186 | 0.378 | 0.429 | 0.133 | 0.281 |
| | $\gamma = 2.5$ | 0.184 | 0.377 | 0.418 | 0.132 | 0.275 |
| | $\gamma = 3.0$ | 0.185 | 0.380 | 0.433 | 0.134 | 0.283 |

Table 8: Ablation study for the number of projections for the ACUS framework on two datasets. The nucleus sampling with $p = 0.7$ is utilized to generate 30 candidate captions for each audio. All sampling methods generate 30 candidate captions and then rerank by the Equation (14).

| Dataset | Number of $L$ | METEOR | ROUGE_L | CIDEr | SPICE | SPIDEr |
|---------|---------------|--------|---------|-------|-------|--------|
| AudioCaps | $L = 10$ | $0.261 \pm 0.001$ | $0.505 \pm 0.002$ | $0.793 \pm 0.008$ | $0.197 \pm 0.001$ | $0.495 \pm 0.005$ |
| | $L = 50$ | $0.262 \pm 0.001$ | $0.509 \pm 0.001$ | $0.807 \pm 0.003$ | $0.192 \pm 0.001$ | $0.5 \pm 0.002$ |
| | $L = 100$ | $0.266 \pm 0.001$ | $0.503 \pm 0.002$ | $0.805 \pm 0.008$ | $0.193 \pm 0.001$ | $0.501 \pm 0.003$ |
| Clotho | $L = 10$ | $0.186 \pm 0.001$ | $0.376 \pm 0.001$ | $0.401 \pm 0.009$ | $0.135 \pm 0.001$ | $0.268 \pm 0.005$ |
| | $L = 50$ | $0.186 \pm 0.001$ | $0.38 \pm 0.001$ | $0.419 \pm 0.004$ | $0.133 \pm 0.001$ | $0.275 \pm 0.003$ |
| | $L = 100$ | $0.187 \pm 0.001$ | $0.382 \pm 0.001$ | $0.42 \pm 0.005$ | $0.134 \pm 0.001$ | $0.275 \pm 0.004$ |

## A.4 QUALITATIVE EXAMPLES

### AUDIOCAPS TEST SET

**Enclap:** Wind blows strongly
**Enclap with contrastive loss:** A motor vehicle engine is running and accelerating
**Enclap with SW:** Wind blowing hard with distant humming of engines

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
