# OpenReview forum: "ACUS: Audio Captioning with Unbiased Sliced Wasserstein Kernel"
_ICLR.cc/2025/Conference — Submitted to ICLR 2025_

### Official Review · Reviewer_4cxP · 2024-10-24

**Soundness:** 2
**Presentation:** 2
**Contribution:** 2
**Rating:** 3
**Confidence:** 4

**Summary:**

This paper is written to solve the exposure bias problem in audio captioning.
They propose the unbiased sliced Wasserstein RBF kernel, which is a better cross-modality similarity measure.
Together with the contrastive learning method, gains are observed in the audio captioning tasks.

**Strengths:**

This paper propose the unbiased sliced Wasserstein kernel framework to improve the audio captioning performance.

**Weaknesses:**

The motivation of the paper does not sound convincing.
Exposure bias is a general problem to auto-regressive networks.
It is not a critical problem for audio captioning.
In general, exposure bias can be mitigated by a better training of model.
I believe using a larger decoder will ease the problem a lot.
Specifically, I do not see in the paper how much the exposure bias problem is harming the audio captioning performance.
Even if we regard this as a serious problem, reinforcement learning (RL) should be a popular way to solve it as RL trains the model according to its inference output. The paper doesn't discuss about RL and address the audio captioning problem in a very narrow perspective.

**Questions:**

As proposing the unbiased kernel is the core technical contribution in the paper, how much gain is there from a biased kernel to a unbiased kernel ?

---

> ### Author Response · Authors · 2024-11-18
> **Response to reviewer 4cxP**
>
> We appreciate the valuable time and feedback of the reviewer to improve your work. We have revised our manuscript and would like to address your concerns outlined below.
>
> **Weakness:** The motivation of the paper does not sound convincing. Exposure bias is a general problem to auto-regressive networks. It is not a critical problem for audio captioning. In general, exposure bias can be mitigated by a better training of model. I believe using a larger decoder will ease the problem a lot. Specifically, I do not see in the paper how much the exposure bias problem is harming the audio captioning performance. Even if we regard this as a serious problem, reinforcement learning (RL) should be a popular way to solve it as RL trains the model according to its inference output. The paper doesn't discuss about RL and address the audio captioning problem in a very narrow perspective.
>
> *Response*: Exposure bias is a critical issue for autoregressive models trained by the maximum likelihood technique, and a lot of studies mentioned clearly how exposure bias harms pretrained model’s performance[1, 2, 3]. Audio captioning models are autoregressive models and are trained by maximum likelihood procedure, thus, they indeed suffer from the exposure bias issue. You mentioned that larger models can solve exposure bias, however, this claim only holds when you have a large amount of training data. There have not been large-scale datasets to train foundation models for audio captioning, thus, it is still crucial to investigate new training and inference techniques for audio captioning.
>
> RL could mitigate the exposure bias issue for autoregressive models, however, it is unstable for training audio captioning. The instability comes from the policy gradient training method, which requires pretraining audio captioning models for a certain number of steps. Thus, the RL method is an ad-hoc method for the audio captioning task, and each model has a unique training process. On the other hand, our method seamlessly utilizes a wide range of encoder-decoder audio captioning models.
>
> Reference:
> [1] Generalization in generation: A closer look at exposure bias.
> [2] Relating Neural Text Degeneration to Exposure Bias.
> [3] Why Exposure Bias Matters: An Imitation Learning Perspective of Error Accumulation in Language Generation.
>
> **Question:** As proposing the unbiased kernel is the core technical contribution in the paper, how much gain is there from a biased kernel to a unbiased kernel?
>
> I would like to provide a comparison between biased and unbiased kernel performance in our framework
>
> | Dataset   | Kernel          | METEOR         | ROUGE_L       | CIDEr         | SPICE         | SPIDEr        |
> |-----------|-----------------|----------------|---------------|---------------|---------------|---------------|
> | AudioCaps | Biased kernel   | 0.255 $\pm$0.002  | 0.495 $\pm$0.002 | 0.782 $\pm$0.004 | 0.184 $\pm$0.003 | 0.485 $\pm$0.003 |
> |           | Unbiased kernel | 0.262 $\pm$0.001  | 0.509 $\pm$0.001 | 0.807 $\pm$0.003 | 0.192 $\pm$0.001 | 0.5 $\pm$0.002   |
> | Clotho    | Biased kernel   | 0.183 $\pm$0.002  | 0.38 $\pm$0.001  | 0.401 $\pm$0.004 | 0.128 $\pm$0.003 | 0.271 $\pm$0.004 |
> |           | Unbiased kernel | 0.186 $\pm$0.001  | 0.38 $\pm$0.001  | 0.419 $\pm$0.004 | 0.133 $\pm$0.001 | 0.275 $\pm$0.003 |
>
> As shown in the above table, our unbiased kernel outperforms the biased kernel significantly in five quantitative evaluation metrics, thus, it demonstrates advantages of the unbiased kernel against the biased kernel in our framework.

---

> > ### Author Response · Authors · 2024-11-25
> > **Look forward to your response**
> >
> > Dear Reviewer 4cxP,
> >
> > We would like to thank you very much for your insightful review, and we hope that our response addresses your previous concerns regarding our paper. However, as the discussion period is expected to end in the next few days, please feel free to let us know if you have any further comments on our work. We would be willing to address any additional concerns you may have. Otherwise, we hope that you will consider increasing your rating.
> >
> > Thank you again for spending time on the paper, we really appreciate it!
> >
> > Best regards,
> >
> > Authors

---

> ### Author Response · Authors · 2024-11-27
> **Reminder of our rebuttals**
>
> Dear Reviewer 4cxP,
>
> We appreciate that you are likely to review multiple other papers, however, as we approach the end of the discussion period. We would greatly appreciate your feedback on our rebuttals. We are happy to address any remaining concerns or questions to improve the paper further. Thank you so much!
>
> Kind regards,
>
> Authors

---

> > ### Comment · Reviewer_4cxP · 2024-11-28
> >
> > Thank you for the author's response.
> >
> > My main concern is that the contrastive method [1], which forms the foundation of your paper, does not appear to be widely adopted. If this method is indeed promising, it should have gained traction among many auto-regressive models, especially decoder-only language models.
> >
> > Given that the original method seems to offer limited contributions, I am apprehensive that the impact of your work might also be constrained.
> >
> > Additionally, could you please provide specific data that illustrates the extent to which exposure bias harms the system, as well as how much of this problem is mitigated by your method?
> >
> > [1] CoNT: Contrastive Neural Text Generation

---

> > > ### Author Response · Authors · 2024-11-28
> > > **Official comment by Authors**
> > >
> > > >My main concern is that the contrastive method [1], which forms the foundation of your paper, does not appear to be widely adopted. If this method is indeed promising, it should have gained traction among many auto-regressive models, especially decoder-only language models.
> > > Given that the original method seems to offer limited contributions, I am apprehensive that the impact of your work might also be constrained.
> > > Additionally, could you please provide specific data that illustrates the extent to which exposure bias harms the system, as well as how much of this problem is mitigated by your method?
> > >
> > > >[1] CoNT: Contrastive Neural Text Generation
> > >
> > > Although our method and [1] are similar in the high-level idea, jointly optimizing a metric with MLE objective and using it at the inference stage, they are two different frameworks. First, we propose a new cross-modality metric based on the USW-RBF kernel, while [1] does not propose a new metric for conditional text generation. Second, our framework does not require a warmup stage to train conditional language models, while [1] requires a warmup stage to train conditional text generation models. Third, in contrast to [1], which requires sampling to produce negative samples for the contrastive method, our proposed framework does not require sampling during the training phase.
> > >
> > > There are two main reasons why [1] is not widely adopted for neural text generation. The first reason is that it requires a warmup stage, which makes the proposed framework in [1] ad hoc and can be challenging to adopt with a wide range of models. The framework in [1] is susceptible to a warmup stage. Each model requires a different warmup stage, therefore, it takes a lot of effort to perform the warmup stage to achieve performance gain. The second reason is that [1] requires sampling negative samples during the training phase, which is a computational burden and increases training time substantially. As mentioned in the previous paragraph, our framework does not suffer from the drawbacks of [1], thereby our framework could be integrated with a wide range of encoder-decoder models, as shown in Table. 2.
> > >
> > > We conducted qualitative experiments in section 5.2 to illustrate that our framework is able to handle exposure bias for the audio captioning task. The experiments show that our framework can improve caption length, lexical diversity, and text-to-audio self-retrieval accuracy. Furthermore, we also provided qualitative examples in Appendix A.4 to demonstrate the superior performance of our framework against MLE and contrastive methods.

---

> > > ### Author Response · Authors · 2024-11-30
> > > **Reminder of our rebuttals (only 1 days left)**
> > >
> > > Dear Reviewer 4cxP
> > >
> > > We appreciate that you are likely to review multiple other papers, however, as we approach the end of the discussion period. We would greatly appreciate your feedback on our rebuttals. We are happy to address any remaining concerns or questions to improve the paper further. Thank you so much!
> > >
> > > Kind regards,
> > >
> > > Authors

---

> > > > ### Author Response · Authors · 2024-12-02
> > > > **Last day to post your response**
> > > >
> > > > Dear Reviewer 4cxP,
> > > >
> > > > We would like to kindly remind you that today is the final day of the discussion period. Given the importance of your feedback to us, we would greatly appreciate it if you could take the time to share your thoughts in the remaining hours.
> > > >
> > > > Thank you for your time and consideration, and we look forward to hearing from you.
> > > >
> > > > Best regards,
> > > >
> > > > The Authors

---

> > > > > ### Comment · Reviewer_4cxP · 2024-12-02
> > > > >
> > > > > Considering the novelty and overall contribution, I am going to maintain my original score for this submission.

---

### Official Review · Reviewer_8m2c · 2024-11-03

**Soundness:** 2
**Presentation:** 3
**Contribution:** 2
**Rating:** 5
**Confidence:** 3

**Summary:**

This paper presents a novel framework tackling the training-inference mismatch in automated audio captioning (AAC) by introducing a temporal-similarity score based on the unbiased sliced Wasserstein RBF (USW-RBF) kernel with rotary positional embeddings. By integrating this score with a stochastic decoding strategy, the approach effectively addresses caption degeneration issues encountered during inference. Experimental results on established AAC benchmark datasets demonstrate notable improvements in model performance, validated through both quantitative and qualitative metrics.

**Strengths:**

1. This paper introduces a novel temporal-similarity score, utilizing the unbiased sliced Wasserstein RBF (USW-RBF) kernel with rotary positional embeddings, to mitigate exposure bias in audio captioning models. Unlike prior research, which has not employed the USW-RBF kernel for cross-modal similarity calculation, this study leverages it to capture temporal dynamics more effectively.
2. The proposed framework is adaptable to a wide range of existing AAC models, with experimental results underscoring its effectiveness in improving model performance.
3. Comprehensive qualitative and quantitative experiments support the method's efficacy. Ablation studies comparing various similarity metrics further highlight the advantages of the USW-RBF kernel over alternative approaches.

**Weaknesses:**

1.	Combining the USW-RBF kernel with stochastic decoding strategies may lead to high computational costs. For example, at inference time, generating $\mathcal{B}$ candidate captions through stochastic decoding results in increased computational time. While the paper demonstrates effectiveness, it lacks a detailed discussion of computational overhead in training and inference.
2.	The performance increase with the proposed approach is relatively minor. According to the original paper, EnCLAP-large achieved SPIDEr scores of 49.5 and 27.8 on AudioCaps and Clotho, while the proposed method reached only 50.0 and 27.5, making the claimed improvement less convincing. According to this comparison, the exposure bias is not that important in AAC tasks.
3.	The application scope of this work appears limited, focusing primarily on AAC tasks. The authors have not explored the framework’s performance on other audio-text multimodal tasks, such as audio-text retrieval, and automatic speech recognition. For instance, can the proposed temporal-similarity score enhance the temporal reasoning capability of the CLAP model?
4.	The study does not deeply explore the sensitivity of the framework to key hyper-parameters, such as the coefficient $\alpha$ in the objective function or the number of Monte Carlo samples $L$ used for the USW-RBF kernel.
5.	The paper dedicates considerable space to explaining the USW-RBF kernel but provides a limited description of how it integrates with the model itself. For example, it’s unclear whether the text embedding is derived from the penultimate layer of the text decoder or from another layer.
6.	Although the paper separates AAC models into encoder-decoder and prefix-tuning architectures, with experiments performed only on the encoder-decoder type. The difference between these two types of architecture is not substantial. Both approaches essentially share the same structure of an audio encoder and a text decoder.
(Minor problems - Line 44: the former architecture → the latter architecture)

**Questions:**

1. Could the authors provide a more detailed explanation of the differences between the two types of AAC architectures? Additionally, could the proposed method be adapted for application within prefix-tuning structures?
2. What is the increase in computational cost introduced by the framework? For example, how much additional inference time is required when using stochastic decoding to generate $\mathcal{B}$ candidate captions?
3. Considering the advanced reasoning and generative capabilities of large language models, frequently used in AAC tasks, could the proposed approach be adapted to work alongside LLMs to achieve higher-quality captions?

---

> ### Author Response · Authors · 2024-11-18
> **Response to reviewer 8m2c (1/2)**
>
> We appreciate the valuable time and feedback of the reviewer to improve your work. We have revised our manuscript and would like to address your concerns outlined below
>
> **Weakness 1:** Combining the USW-RBF kernel with stochastic decoding strategies may lead to high computational costs
>
> We understand the reviewer's concern regarding the trade-off between efficiency and effectiveness. Thus, we would like to provide a comparison of inference time between our method and the traditional approaches. The inference time is measured using the real-time-factor (RTF) metric, which measures how many seconds a method needs to generate a caption.
>
> | Method     | Inference time on a A6000 GPU (RTF) |
> |------------|-------------------------------------|
> | MLE        | 0.33                                |
> | MLE + CL   | 0.65                                |
> | MLE + ACUS | 0.81                                |
>
> As illustrated in the above table, our framework is able to generate audio captions real time, therefore, it can be utilized for real-world applications. The inference time of our method increases twice compared with the baseline method, however, our method is able to generate more diverse and high-quality captions.
>
> We also understand the reviewer's concern about the computational cost of the Monte Carlo sampling during training and inference. However, sampling projecting directions for sliced Wasserstein is very computationally efficient i.e., it is equivalent to sampling from a standard multivariate Gaussian. Moreover, projecting directions do not need to be resampled multiple times i.e., we can reuse the projecting directions for some iterations before resampling. Finally, from Proposition 3, the approximation rate is $\mathcal{O}(L^{-1/2})$ which suggests that we might not need a very large number of Monte Carlo samples to obtain a good approximation. According to empirical experiments, we choose the number of Monte Carlo samples $L=50$ for both training and inference in our framework. Furthermore, We conducted an ablation study on the trade-off between computational cost and the quality of generated captions at the inference stage in the Table. 8.
>
> **Weakness 2:** The performance increase with the proposed approach is relatively minor.
>
> We tried to reproduce the experimental results reported in the EnCLAP backbone by using author’s public codebase, however, the reproduced results are slightly lower than the original results. We realized that the mismatch in terms of results was caused by the number of GPUs used to train the EnCLAP model. The authors used four A100 GPUs to achieve the reported results, while we used two A100 GPUs to reproduce the results. Therefore, we reported the reproduced results in our manuscript for a fair comparison. With the same training setups, the improvement of our method is substantially significant compared with the baseline EnCLAP-large with the MLE method. The SPIDEr score improves from 0.48 to 0.5 for the AudioCaps dataset and from 0.273 to 0.275 for the Clotho dataset.
>
> We realized that the quantitative evaluation metrics are not good enough to evaluate the performance of the text generation task, audio caption generation. Thus, we carried out comprehensive qualitative experiments. Both subjective and objective metrics are used to evaluate our method and baselines in the section 3.2. According to our qualitative experiments, our method is able to enhance caption length, lexical diversity, and descriptiveness of audio captions.
>
> **Weakness 3:**  The application scope of this work appears limited.
>
> We develop the ACUS framework to deal with exposure bias and local temporal distortion when measuring cross-modal alignment of the audio caption tasks. The audio-text retrieval and speech recognition tasks are out of the scope of our study. Furthermore, the speech recognition task is not an ideal application for our proposed framework. The speech recognition task requires measuring monotonic alignment between acoustic and linguistic features, and our method is not better than traditional methods such as Dynamic Time Wrapping. Regarding audio-text retrieval, we do believe that our framework can enhance audio-text retrieval accuracy, however, it is challenging to compare with the pretrained CLAP model. The reason is that we can not acquire a large number of GPUs to pretrain the CLAP backbone using our proposed framework on a large amount of datasets. Thus, we are not able to conduct a fair comparison between our framework and the contrastive learning for the CLAP backbone on audio-text retrieval.

---

> ### Author Response · Authors · 2024-11-18
> **Response to reviewer 8m2c (2/2)**
>
> **Weakness 4:** The study does not deeply explore the sensitivity of the framework to key hyper-parameters
>
> We first clarify that the coefficient $\alpha$ in equation 14 is the hyper-parameter for the inference stage. The training objective is equation 13, in which there is no coefficient $\alpha$. During training, the coefficient’s ratio between likelihood and USW-RBF terms equals 1, thereby, the coefficient $\alpha$ is set to 0.5 to be compatible with the training stage. Although tuning the coefficient $\alpha$ during inference can increase slightly the performance, it is trivial to conduct hyper-parameter tuning for coefficient $\alpha$. We provided the ablation study for the number of Mote Carlo samples $L$ in Table.8 in Appendix A.3.
>
> **Weakness 5:** The paper dedicates considerable space to explaining the USW-RBF kernel but provides a limited description of how it integrates with the model itself.
>
> Thank the reviewer for helping us improve the clarity of our submission. We use the text embedding from the final hidden layer of the text decoder to measure the cross-modal similarity between audio and captions. We will update the manuscript according to your comments to explain our method more clearly. We detailed the training and inference stages of our proposed framework in section 3.2. We also demonstrated how the proposed kernel is integrated with the encoder-decoder architecture for training and inference with audio captioning models in Figure. 1.
>
> **Weakness 6 and question 1:**  Although the paper separates AAC models into encoder-decoder and prefix-tuning architectures, with experiments performed only on the encoder-decoder type. Could the proposed method be adapted for application within prefix-tuning structures?
>
> Thank the reviewer for helping point out the typo. Although both encoder-decoder and prefix-tuning architectures have an audio encoder and text decoder, they are different backbones for the audio captioning task. Both the encoder and the decoder of encoder-decoder architecture are finetuned during training, while only the encoder of prefix-tunning architecture is finetuned during training. Furthermore, the generation at the inference stage is different between the two architectures. Regarding the encoder-decoder architecture, the encoder first extracts semantic representation from a given audio and then sequentially generates an audio caption based on the semantic representation. Regarding prefix-tuning architecture, the encoder maps a given audio to a prefix on the text embedding space of the fixed-pretrained text decoder. Then, the text decoder sequentially generates an audio caption based on the prefix.
>
> We did experiments on the prefix-tuning backbones for our method, but our method does not work on prefix-tuning backbones. The explanation is that the encoder of prefix-tuning aims to learn the mapping from audio features to text embedding, but the encoder is not able to learn meaningful representation for acoustic features. Thus, it is challenging for the USW-RBF to measure the similarity across acoustic and linguistic modalities precisely. As a result, we emphasized in our manuscript that our framework is merely developed for encoder-decoder architecture.
>
> **Question 2:**  What is the increase in computational cost introduced by the framework?
>
> We understand the reviewer's concern regarding the trade-off between efficiency and effectiveness. Thus, we would like to provide an experiment regarding using stochastic decoding methods to generate $B$ candidate captions in our framework. The inference time is measured by using the real-time-factor (RTF) metric, which measures how many seconds a method needs to generate a caption.
>
> | Method(top-p=0.7) | Inference time on a A6000 GPU (RTF) |
> |-------------------|-------------------------------------|
> | ACUS with $B=10$  | 0.55                                |
> | ACUS with $B=20$  | 0.72                                |
> | ACUS with $B=30$  | 0.81                                |
>
> As illustrated in the above table, our framework can generate audio captions in real-time, which makes it suitable for real-world applications.
>
> **Question 3:** Could the proposed approach be adapted to work alongside LLMs to achieve higher-quality captions?
>
> I do believe that if we leverage advanced LLMs for AAC using our framework, there will be a significant improvement in the AAC task. However, LLMs should be integrated for the AAC task in the encoder-decoder scheme to be compatible with our framework rather than the prefix-tuning scheme.

---

### Official Review · Reviewer_bPjz · 2024-11-03

**Soundness:** 2
**Presentation:** 2
**Contribution:** 2
**Rating:** 6
**Confidence:** 4

**Summary:**

The paper proposes a new framework for audio captioning designed to mitigate exposure bias and improve temporal cross-modal similarity measurement. The authors claim that traditional audio captioning models trained via maximum likelihood estimation face exposure bias, leading to "degeneration" in generated captions. This paper introduces the unbiased sliced Wasserstein (USW-RBF) kernel equipped with rotary positional embedding to capture temporal information across acoustic and linguistic modalities, thereby reducing exposure bias. The authors show improvements on benchmark datasets (AudioCaps and Clotho)

**Strengths:**

- The paper is decently written and easy to follow for an expert. However, I would like to mention it might be difficult for a traditional audio captioning community to read. A bit more background on the 1, the unbiased sliced Wasserstein RBF kernel would have been appreciated.  The equations could have been improved by defining the notations better. Fro examples, if I want to read Eqtn 9., I need to find what the notations mean somewhere else in the paper.
- The problem handled in the paper is new. The approach is also novel. ACUS combines the USW-RBF kernel with stochastic decoding methods like nucleus and top-k sampling to alleviate exposure bias during inference. A lot of work in audio captioning ideally propose new architectures. This paper brings a fresh perspective in the problem space.
- The evaluation is sound. The 2 usual benchmark datasets are used and it is also combined with human evaluation. The metrics of descriptiveness, correctness, and fluency are good metrics for comparison as ideal benchmark metrics seem to have saturated and require style memorization.

**Weaknesses:**

- The abstract says" "Prior works propose the contrastive method to deal with caption degeneration. However, the contrastive method ignores the temporal information when measuring similarity across acoustic and linguistic modalities, leading to inferior performance." -- which contrastive method and how does it ignore the "the temporal information when measuring similarity across acoustic and linguistic modalities"? This first line of the paper is very difficult to understand.
- The Monte Carlo sampling and stochastic gradient optimization may increase computational costs, potentially impacting efficiency in real-world large-scale applications.
- While I understand that the authors focus on Enc-Dec framework, a good number of baselines were missed for evaluation. ACUS can act as complimentary to most other methods proposed in literature as all methods require an audio encoder and a language decoder (including prefix based architectures). Thus, some baselines were missed. See [1,2] as examples and papers compared in [1,2].
- The analysis section is just ablations. A deeper analysis section (see questions below) would have strengthened the paper.


### Citations

[1] https://ieeexplore.ieee.org/abstract/document/10448030.
[2] https://ieeexplore.ieee.org/abstract/document/10096877.

**Questions:**

- How does the performance of USW-RBF compare with other non-Wasserstein-based kernels for audio captioning tasks?
- What are the potential trade-offs between the accuracy improvements and the computational costs introduced by Monte Carlo sampling and stochastic gradient optimization in ACUS?
- Why was the rotary positional embedding favored over other encoding techniques, and could alternative embeddings further enhance the results?
- Why audio captioning? Can the method be useful to other tasks? Audio understanding? Speech Recognition?

---

> ### Author Response · Authors · 2024-11-14
> **Response to reviewer bPjz (1/2)**
>
> We appreciate the valuable time and feedback of reviewer to improve your work. We have revised our manuscript and would like to address your concerns outlined below
>
> **Weakness 1:** which contrastive method and how does it ignore "the temporal information when measuring similarity across acoustic and linguistic modalities"?
>
> We would like to thank the reviewer for helping us improve the quality of the submission. We will definitely rephrase these sentences to make our abstract more clear . We mention the classical contrastive learning (CL) method for multimodal learning in this work. The relevant CL works for audio captioning are cited in our introduction for clarity. The main drawback of CL method is performing an average-pooling operation before measuring the cosine similarity across acoustic and linguistic modalities. Thus, the temporal information in acoustic and linguistic features are discarded due to the average-pooling operation. Our method avoids this drawback by directly measuring the discrepancy between two sequences.
>
> **Weakness 2:** The Monte Carlo sampling and stochastic gradient optimization may increase computational costs.
>
> Thank you for your insightful comments. We understand the reviewer's concern about the computational cost of the Monte Carlo sampling. However, sampling projecting directions for sliced Wasserstein is very computationally efficient i.e., it is equivalent to sampling from a standard multivariate Gaussian. Moreover, projecting directions do not need to be resampled multiple times i.e., we can reuse the projecting directions for some iterations before resampling. Finally, from Proposition 3, the approximation rate is $\mathcal{O}(L^{-1/2})$ which suggests that we might not need a very large number of Monte Carlo samples to obtain a good approximation.
>
> **Weakness 3:** missing prefix-tuning baselines for the audio captioning task
>
> Thank you for pointing out the relevant prefix-tuning baselines for audio captioning. We will update the experiment section to include all relevant baselines based on your suggestion. Also, we would like to emphasize that our proposed method still significantly outperforms all prefix-tuning baselines for audio captioning.
>
> | Dataset   | Method      | METEOR       | ROUGE_L      | CIDEr        | SPICE        | SPIDEr       |
> |-----------|-------------|--------------|--------------|--------------|--------------|--------------|
> | AudioCaps | [1]         | 0.256        | 0.525        | 0.751        | 0.186        | 0.471        |
> |           | [2]         | 0.240        | 0.503        | 0.733        | 0.177        | 0.455        |
> |           | ACUS (ours) | 0.262 $\pm$ 0.001 | 0.509 $\pm$0.001 | 0.807 $\pm$0.003 | 0.192 $\pm$0.001 | 0.5 $\pm$0.002 |
> | Clotho    | [1]         | 0.177        | 0.395        | 0.411        | 0.125        | 0.224        |
> |           | [2]         | 0.170        | 0.378        | 0.392        | 0.118        | 0.255        |
> |           | ACUS (ours) | 0.186$\pm$0.001 | 0.38$\pm$0.001  | 0.419$\pm$0.004 | 0.133$\pm$0.001 | 0.275$\pm$0.003 |
>
> Citations
>
> [1] https://ieeexplore.ieee.org/abstract/document/10448030.
>
> [2] https://ieeexplore.ieee.org/abstract/document/10096877.

---

> ### Author Response · Authors · 2024-11-15
> **Response to reviewer bPjz (2/2)**
>
> **Question 1:** How does the performance of USW-RBF compare with other non-Wasserstein-based kernels for audio captioning tasks?
>
> We compare with soft-DTW and Dynamic Time Wrapping (DTW) methods which are equivalent to non-Wasserstein kernels as mentioned in [1,2]. The experimental results are illustrated in the Table. 5. As discussed thoroughly in our work, DTW and soft-DTW are incapable of handling local temporal distortion for the audio captioning task, while our kernel is able to deal with the distortion. Thus, our proposed kernel substantially outperforms DTW and soft-DTW for measuring similarity across acoustic and linguistic modalities for the audio captioning task.
>
> References:
>
> [1] A kernel for time series based on global alignments
>
> [2] Fast Global Alignment Kernels
>
> **Question 2:** What are the potential trade-offs between the accuracy improvements and the computational costs introduced by Monte Carlo sampling and stochastic gradient optimization in ACUS?
>
> We conducted an ablation study regarding trade-offs between efficiency and effectiveness of our method regarding Monte Carlo sampling in the Table. 8 in the Appendix.3.  According to proposition 3, the approximation rate of USW-RBF kernel is $\mathcal{O}(L^{-1/2})$ which suggests that we might not need a very large number of Monte Carlo samples to obtain a good approximation. As shown in the Table. 8, our proposed kernel can improve the performance of the baseline method significantly with a decent number of Monte Carlo samples $L=50$. I would like to illustrate the experimental results below for clarity:
>
> | Dataset   | Method               | METEOR       | ROUGE_L      | CIDEr        | SPICE        | SPIDEr       |
> |-----------|----------------------|--------------|--------------|--------------|--------------|--------------|
> | AudioCaps | Enclap + MLE         | 0.254        | 0.5          | 0.77         | 0.186        | 0.48         |
> |           | Enclap + ACUS (L=50) | 0.262$\pm$0.001 | 0.509$\pm$0.001 | 0.807$\pm$0.003 | 0.192$\pm$0.001 | 0.5$\pm$0.002  |
> | Clotho    | Enclap + MLE         | 0.182        | 0.38         | 0.417        | 0.13         | 0.273        |
> |           | Enclap + ACUS (L=50) | 0.186$\pm$0.001 | 0.38$\pm$0.001 | 0.419$\pm$0.004 | 0.133$\pm$0.001 | 0.275$\pm$0.003 |
>
> **Question 3:**
>
> We choose rotary positional embedding based on the ablation study in the table. 6. There are a lot of the same observations[1, 2] aligned with our ablation study, which is the rotary PE outperforming absolute PE in encoding positional information for sequential data. We believe that if we have a better positional encoding(PE) technique, the PE technique can enhance our proposed method.
>
> References:
>
> [1] The llama 3 herd of models
>
> [2] Palm: Scaling language modeling with pathways
>
> **Question 4:** Why audio captioning? Can the method be useful to other tasks? Audio understanding? Speech Recognition?
>
> We chose the audio captioning task to show the effectiveness of our proposed framework for two reasons. The first reason is that the audio caption task requires cross-modal alignment, in which temporal information from each modality is crucial to consider when measuring cross-modal similarity between acoustic and linguistic features to generate high-quality captions. The second reason is that the cross-modal alignment for the audio captioning task is not a monotonic alignment. Thus, the traditional method, like Dynamic Time Wrapping (DTW), can not handle non-monotonic alignments since local temporal distortion exists.
>
> The audio understanding and audio question-answering tasks are similar to the audio captioning task. All of them require measuring cross-modal alignment to better generate either a caption or answer for a given audio, therefore, our method is well-suited for these tasks. I do not see any benefits in applying our method to the speech recognition task. The speech recognition task requires measuring monotonic alignment between acoustic and linguistic features, and our method is not better than traditional methods, such as DTW, for handling monotonic alignments.

---

> ### Author Response · Authors · 2024-11-20
> **Looking forward to your response**
>
> As the rebuttal period is coming to an end, we hope to have clarified the novelty of our contribution and addressed the reviewer's concerns. We truly appreciate the reviewer's time on our paper and are looking forward to your responses and/or any other potential questions. Your feedback would be very helpful for us to identify any ambiguities in the paper for further improvement.

---

> > ### Comment · Reviewer_bPjz · 2024-11-21
> > **Thank You for your response**
> >
> > Thank You for your response and in the effort taken in the rebuttal.
> >
> > While the rebuttal addressed most of my concerns and can vouch for the soundness of the approach, I am overall not very satisfied with the complexity of the proposed approach for solving a task as simple as audio captioning. A task that requires just acoustic element recognition may be better solved by improved audio perception (better encoders). Additionally, in my experience, incremental score improvements on AudioCaps and Clotho dont say much and might only reflect style memorization (current evaluation metrics are not the best to judge -- incase of human judgments, the sample taken is rather small and incase it was done on mturk, mturk evaluators are not very strong audio evaluators -- more details on human evaluation setup would also be appreciated) While LALMs have been on the rise, some problems may be solved at scale and this problem might be more prominent in audio captioning due to the current small scale models used for audio captioning. I also resonate with some issues addressed by other reviewers.
> >
> > I am increasing my score to 6 for the effort taken as I believe the authors have answered my questions/doubts, which was the reason behind my earlier score and the point of a rebuttal.

---

> > > ### Author Response · Authors · 2024-11-30
> > > **Thank you for your fruitful feedback**
> > >
> > > Thank you for your constructive feedback to help us improve our work!

---

### Official Review · Reviewer_2JoP · 2024-11-03

**Soundness:** 2
**Presentation:** 2
**Contribution:** 2
**Rating:** 3
**Confidence:** 4

**Summary:**

This paper introduces a novel approach for audio captioning to address issues related to exposure bias and temporal misalignment. Here’s a summary of the key contributions:
1. Unbiased Sliced Wasserstein RBF Kernel (USW-RBF): The authors propose a novel kernel method to accurately measure cross-modal similarity between audio and textual data. This kernel, equipped with rotary positional embedding, captures temporal information more effectively than traditional methods, addressing limitations like dimensionality and temporal distortion.
2. Mitigating Exposure Bias: ACUS employs stochastic decoding techniques, such as nucleus and top-k sampling, to reduce exposure bias during the inference stage, enhancing caption diversity and quality. This is achieved by leveraging the USW-RBF kernel to improve alignment between generated captions and audio inputs.
3. Extensive Evaluation and Compatibility: The framework’s efficacy is validated through experiments on two datasets, AudioCaps and Clotho. Results demonstrate improved caption length, lexical diversity, and self-retrieval accuracy, and compatibility with diverse encoder-decoder architectures.

In essence, ACUS represents an advancement in audio captioning by integrating the unbiased USW-RBF kernel with stochastic decoding, leading to more descriptive and temporally coherent audio captions.

**Strengths:**

1. The introduction of the unbiased sliced Wasserstein RBF (USW-RBF) kernel, which captures temporal information across audio and text modalities, is an advancement. By accounting for temporal alignment, it addresses limitations in prior contrastive methods that often ignore the temporal structure of audio data.

2. ACUS effectively addresses exposure bias—a common issue in captioning tasks—by combining the USW-RBF kernel with stochastic decoding methods. This approach ensures that generated captions maintain diversity and relevance across varying contexts.

3. The ACUS framework enhances not only the length and diversity of captions but also their semantic alignment with audio events. By capturing temporal details, it generates more descriptive and meaningful captions, which are validated by both quantitative metrics and qualitative assessments.

4. The paper thoroughly derives and proves the properties of the USW-RBF kernel, reinforcing its validity for multimodal tasks. Additionally, by introducing a practical approach to reducing exposure bias, it offers a methodological contribution that may extend beyond audio captioning to other sequence generation tasks.

**Weaknesses:**

1. While the paper introduces a promising method, the improvements observed in the current experiments are relatively modest. To more rigorously validate the effectiveness of your approach, I recommend evaluating your model on additional, more challenging benchmarks, which may make this work more convincing if your method reach higher score in these benchmarks, such as:

a. SUPERB: This benchmark includes a broad range of speech processing tasks, covering content, speaker, semantics, and paralinguistics. It would provide a comprehensive baseline, helping clarify how well your model generalizes across core speech tasks.

b. Dynamic-SUPERB: This benchmark extends SUPERB with instruction-tuning and zero-shot tasks, pushing models to handle more complex and varied speech processing scenarios. Testing on Dynamic-SUPERB could demonstrate your method’s robustness and adaptability in handling multi-task and instruction-following requirements, offering deeper insights into its generalization capabilities.

c. SpeechCaps: Given the emphasis in your work on speaker-specific and temporal information, SpeechCaps offers a relevant test for multi-talker and speaking style captioning. Its focus on speaker and prosodic information could highlight the strengths of your model in more intricate, real-world audio scenarios, such as multi-speaker dialogues and expressive speech.

2. Authors provide a detailed explanation of the USW-RBF kernel. However, it lacks sufficient details on how this kernel is integrated within the overall model architecture. You can try these to make it better, such as:

a. Integration Details: Please provide a clearer, step-by-step description of how the USW-RBF kernel is incorporated into the model pipeline.

b. Diagram or Flowchart: Consider adding a diagram or flowchart that visualizes the integration process, illustrating where and how the USW-RBF kernel interacts with audio and textual embeddings within the architecture.

**Questions:**

1. There are some minor errors, like in line 187, it should be \( \nu = \frac{1}{N} \sum_{j=0}^N \delta_{z_y^j} \), not \( \nu = \frac{1}{M} \sum_{j=0}^N \delta_{z_y^j} \), to make two empirical distributions have the same number of supports. I did not thoroughly inspect every math part in this paper, but I think authors could check the whole paper again thoroughly.  Also, in conclusion, it should be "unbiased kernel", not "unbias kernel". (No worries, they are just minor error, but it is better to correct for clarity.)

2. As I mentioned in weakness part, the reported improvements over baselines appear modest. Could you provide more analysis on how the proposed method performs in more challenging scenarios (e.g., multi-speaker or noisy environments) to better highlight its strengths? If not, do you believe that temporal information is important in audio captioning task?

3. The application area of this work seems limited. Do you have any plans to extend this work for multilingual audio captioning or automatic speech recognition? If so, how might the kernel method adapt to language diversity in audio processing? how you modify the text embedding under multilingual scenario?

4. Since you use stochastic decoding strategies in inference stage, which may lead to high computational costs, and the reported score in your results is not very decent, we might not need such a high-cost method to get a minor improvement. Thus, could you provide more details on the differences in diversity and quality of captions generated by your approach?

---

> ### Author Response · Authors · 2024-11-15
> **Response to reviewer 2JoP (1/2)**
>
> We appreciate the valuable time and feedback of reviewer to improve your work. We have revised our manuscript and would like to address your concerns outlined below
>
> Before addressing the review's concerns, we would like to emphasize the main contribution of our work is that *we develop a framework method to address the exposure bias of current training frameworks, maximum likelihood and contrastive learning, for audio captioning models. Our contribution is a new framework rather than a new model for the audio captioning task*. We dedicated a section 3.2 and the Figure. 1 to detail the training and inference procedure of our proposed framework. The Figure.1 demonstrates how the proposed USW-RBF kernel is integrated into encoder-decoder architectures for training and inference stages. Furthermore, we would like to argue that audio and speech are similar to some extent, but they are different. Speech tasks focus on understanding the linguistic and speaker information of audio. On the other hand, audio tasks focus on understanding acoustic information and sound events of audio.
>
> **Weakness 1:** Evaluate our method on speech benchmarks: SUPERB, Dynamic-SUPERB, and SpeechCaps
>
> All those mentioned benchmarks are designed to evaluate the transferable capability of pretrained speech models which is totally irrelevant to our work, our work mainly focuses on solving the exposure bias for audio captioning. Even though speech and audio are similar to some extent, they are two different. I would like to articulate the reasons why these benchmarks are not suitable for evaluating our proposed method as follows. Both SUPERB and Dynamic-SUPERB are used to evaluate the transferable capability of universal pretrained speech models, which is irrelevant to the main research question of our work, solving the exposure bias issue for audio captioning models. The second reason is that we propose a new framework to deal with the exposure bias of audio captioning tasks rather than a new model or architecture for transfer learning, therefore, there is no connection between our work and SUPERB and Dynamic-SUPERB benchmarks. The SpeechCaps benchmark is designed to evaluate the understanding capabilities of speech models in prosodic information. Thus, it is not appropriate to use for benchmarking our method.
>
> **weakness 2:** lack of sufficient details on how this kernel is integrated within the overall model architecture
>
> We detailed the training and inference stages of our proposed framework in the section 3.2. We also demonstrated how the proposed kernel is integrated with the encoder-decoder architecture for training and inference with audio captioning models in the Figure. 1.

---

> ### Author Response · Authors · 2024-11-15
> **Response to reviewer 2JoP (2/2)**
>
> **Question 1:** There are some minor errors, like in line 187, it should be ( \nu = \frac{1}{N} \sum_{j=0}^N \delta_{z_y^j} ), not ( \nu = \frac{1}{M} \sum_{j=0}^N \delta_{z_y^j} ), to make two empirical distributions have the same number of supports.
>
> We would like to thank the reviewer for pointing out these typos to improve our submission. We will update our manuscript accordingly based on your suggestion.
>
> **Question 2:** Could you provide more analysis on how the proposed method performs in more challenging scenarios (e.g., multi-speaker or noisy environments) to better highlight its strengths? If not, do you believe that temporal information is important in audio captioning task?
>
> Those mentioned benchmarks and settings are irrelevant to our work, therefore, we do not think our proposed method should be evaluated on the suggested benchmarks. The explanations are articulated in my aforementioned response.
> We conducted experiments to compare our proposed method against the contrastive learning and baseline methods, which are incapable of considering temporal information. Our method remarkably outperforms baseline methods in terms of quantitative and qualitative metrics for the audio captioning task in the section 5.1 and 5.2, respectively. We also provide qualitative examples in Appendix. A4 to demonstrate that our method is capable of generating more high-quality captions against baseline methods. According to experimental results, the temporal information is crucial for the audio captioning task, therefore, it should be taken into account when measuring the similarity between acoustic and linguistic modalities for encoder-decoder audio captioning models.
>
> **Question 3:**  The application area of this work seems limited.
>
> I think that applying our method for multilingual audio captioning would be great. If there is a standard dataset for multilingual audio captioning, we would like to evaluate our method on that dataset to show its effectiveness. There is no benefit to utilizing our method for the automatic speech recognition (ASR) task. We developed the ACUS framework to deal with the local temporal distortion issue for text-audio alignment, and the ASR task does not suffer from this issue. The reason is that ASR task requires a monotonic alignment across acoustic features and ground-truth transcript, therefore there is no benefit of leveraging our method for ASR.
>
> Our proposed kernel method is a non-parametric method and works on the output space of the audio encoder and text decoder. Thus, our method can be seamlessly utilized with the multilingual audio captioning tasks for encoder-decoder models without any modification in model's architectures.
>
> **Question 4:**  Since you use stochastic decoding strategies in the inference stage, which may lead to high computational costs, and the reported score in your results is not very decent, we might not need such a high-cost method to get a minor improvement. Thus, could you provide more details on the differences in diversity and quality of captions generated by your approach?
>
> We understand the reviewer's concern regarding the trade-off between efficiency and effectiveness. Thus, we would like to provide a comparison of inference time between our method and the traditional approaches. The inference time is measured by using real-time-factor (RTF) metric, which measures how many seconds a method needs to generate a caption.
>
> | Method     | Inference time on a A6000 GPU (RTF) |
> |------------|----------------------------|
> | MLE        | 0.33                                |
> | MLE + contrastive learning   | 0.65                            |
> | MLE + ACUS (ours) | 0.81                         |
>
> As illustrated in the above table, our framework is able to generate audio captions real time, therefore, it can be utilized for real-world applications. The inference time of our method increases twice compared with the baseline method, however, our method is able to generate more diverse and high-quality captions. We conduct extensive qualitative experiments in the Section 5.2(both subjective and objective experiments)  to demonstrate the high-quality of generated captions by leveraging our proposed framework against traditional methods. Our method can improve caption length, lexical diversity, and semantic of generated captions, as shown in the Table. 3, and the ACUS framework substantially enhances the descriptiveness and correctness of generated caption, evaluated by human evaluation, in the Table. 4. Furthermore, we also provide qualitative examples in the Appendix. A4.

---

> > ### Author Response · Authors · 2024-11-25
> > **Look forward to your response**
> >
> > Dear Reviewer 2JoP,
> >
> > We would like to thank you very much for your insightful review, and we hope that our response addresses your previous concerns regarding our paper. However, as the discussion period is expected to end in the next few days, please feel free to let us know if you have any further comments on our work. We would be willing to address any additional concerns you may have. Otherwise, we hope that you will consider increasing your rating.
> >
> > Thank you again for spending time on the paper, we really appreciate it!
> >
> > Best regards,
> >
> > Authors

---

> ### Author Response · Authors · 2024-11-27
> **Reminder of our rebuttals**
>
> Dear Reviewer 2JoP,
>
> We appreciate that you are likely to review multiple other papers, however, as we approach the end of the discussion period. We would greatly appreciate your feedback on our rebuttals. We are happy to address any remaining concerns or questions to improve the paper further. Thank you so much!
>
> Kind regards,
>
> Authors

---

> ### Author Response · Authors · 2024-11-28
> **Reminder of our rebuttals (only 3 days left)**
>
> Dear Reviewer 2JoP,
>
> We appreciate that you are likely to review multiple other papers, however, as we approach the end of the discussion period. We would greatly appreciate your feedback on our rebuttals. We are happy to address any remaining concerns or questions to improve the paper further. Thank you so much!
>
> Kind regards,
>
> Authors

---

> ### Author Response · Authors · 2024-11-29
> **Reminder of our rebuttals (only 2 days left)**
>
> Dear Reviewer 2JoP,
>
> We appreciate that you are likely to review multiple other papers, however, as we approach the end of the discussion period. We would greatly appreciate your feedback on our rebuttals. We are happy to address any remaining concerns or questions to improve the paper further. Thank you so much!
>
> Kind regards,
>
> Authors

---

> > ### Author Response · Authors · 2024-11-30
> > **Reminder of our rebuttals (only 1 days left)**
> >
> > Dear Reviewer 2JoP,
> >
> > It has been two weeks since we uploaded our rebuttals. We appreciate that you are likely to review multiple other papers, however, as we approach the end of the discussion period. We would greatly appreciate your feedback on our rebuttals. We are happy to address any remaining concerns or questions to improve the paper further. Thank you so much!
> >
> > Kind regards,
> >
> > Authors

---

> > > ### Author Response · Authors · 2024-12-02
> > > **Last day to post your response**
> > >
> > > Dear Reviewer 2JoP,
> > >
> > > We would like to kindly remind you that **today is the final day of the discussion period**, and we have not yet received your feedback since November 23rd. We have put a large effort into addressing your concerns. Given the importance of your feedback to us, we would greatly appreciate it if you could take the time to share your thoughts in the remaining hours.
> > >
> > > Thank you for your time and consideration, and we look forward to hearing from you.
> > >
> > > Best regards,
> > >
> > > The Authors

---

### Official Review · Reviewer_fKkk · 2024-11-04

**Soundness:** 4
**Presentation:** 4
**Contribution:** 3
**Rating:** 8
**Confidence:** 4

**Summary:**

This paper introduces ACUS (Audio Captioning with Unbiased Sliced Wasserstein kernel), a novel framework that addresses exposure bias and temporal misalignment issues in audio captioning systems. The key technical contribution is the development of an unbiased sliced Wasserstein RBF (USW-RBF) kernel equipped with rotary positional embeddings, which effectively measures similarity between acoustic and linguistic features while preserving temporal information. Experimental results on AudioCaps and Clotho datasets demonstrate significant improvements over state-of-the-art methods across multiple metrics.

**Strengths:**

The paper introduces an unbiased sliced Wasserstein RBF (USW-RBF) kernel that effectively handles temporal information across modalities while avoiding dimensionality curse issues that affect traditional Wasserstein distances.

Strong Theoretical Foundation: Provides formal proofs for the kernel's properties (positive definiteness, unbiasedness).
Demonstrates convergence rate for Monte Carlo estimation.

Comprehensive Evaluation: Tests on multiple datasets (AudioCaps and Clotho). Uses both automatic metrics and human evaluation
Includes detailed ablation studies for various components.

Achieves state-of-the-art performance on multiple metrics

**Weaknesses:**

No analysis of computational overhead from the USW-RBF kernel

Unclear how the method performs on longer audio sequences

While ablation studies are included, there's limited discussion of how sensitive the method is to various hyperparameters
Could benefit from more guidance on hyperparameter selection for new datasets.

Lacks detailed analysis of failure cases

**Questions:**

How does the computational complexity scale with audio length and batch size compared to baseline methods?
How robust is the method to different audio qualities or noise levels? Was this tested?
What is the impact of different positional embedding choices on the final performance? While rotary embeddings performed best, is there a theoretical justification for this?

---

> ### Author Response · Authors · 2024-11-20
> **Response to reviewer fKkk**
>
> We appreciate the valuable time and feedback of the reviewer to improve your work. We have revised our manuscript and would like to address your concerns outlined below.
>
> **Weakness 1:** No analysis of computational overhead from the USW-RBF kernel
>
> We understand the reviewer's concern regarding the computational overhead of our kernel. Our kernel is approximated using the Monte Carlo method, however, sampling projecting directions for sliced Wasserstein is very computationally efficient i.e., it is equivalent to sampling from a standard multivariate Gaussian. Moreover, projecting directions do not need to be resampled multiple times i.e., we can reuse the projecting directions for some iterations before resampling. Finally, from Proposition 3, the approximation rate is $\mathcal{O}(L^{-1/2})$ which suggests that we might not need a very large number of Monte Carlo samples to obtain a good approximation. We conducted the trade-off between efficiency and effectiveness of our framework in Table 8 in Appendix A.3.
>
> **Weakness 2:** Unclear how the method performs on longer audio sequences
>
> Our proposed framework does not distinguish between handling short and long audio. Technically, our method measures similarity across acoustic and linguistic modalities similarly, regardless of audio duration.
>
> **Weakness 3:** While ablation studies are included, there's limited discussion of how sensitive the method is to various hyperparameters. Could benefit from more guidance on hyperparameter selection for new datasets.
>
> We conducted the hyperparameter selection for our method on the number of Monte Carlo samples $L$ and the bandwidth parameter $\gamma$ in Table.7 and Table.8 in Appendix A3, respectively.
>
> **Weakness 4:** Lacks detailed analysis of failure cases
>
> We would like to thank the reviewer for this insightful feedback. We will incorporate an analysis of failure cases in the Appendix.
>
> **Question 1:** How does the computational complexity scale with audio length and batch size compared to baseline methods?
>
> The baseline method, contrastive learning method, has a computational complexity of $\mathcal{O}(n)$ for a minibatch of $n$ audio-caption pairs, where each audio and caption has a length of $d$. While our proposed method has a computational complexity of $\mathcal{O}(n.d.\log{d})$
>
> **Question 2:** How robust is the method to different audio qualities or noise levels? Was this tested?
>
> Since the robustness of our method for noisy audio is not the primary objective of our work, we have not tested it in noisy audio.
>
> **Question 3:** What is the impact of different positional embedding choices on the final performance? While rotary embeddings performed best, is there a theoretical justification for this?
>
> We conducted the ablation study on the impact of positional embedding (PE) techniques on our framewok in Table. 6. The rotary PE technique achieves the highest performance based on the empirical experiment. We hypothesize that the rotary PE technique is well-suited for our framework because it effectively encodes the positional information for a long sequence of audio features.

---

> > ### Comment · Reviewer_fKkk · 2024-11-26
> >
> > Thank you for your clarifications.

---

> > > ### Author Response · Authors · 2024-11-26
> > > **Thank You**
> > >
> > > Dear Reviewer fKkk,
> > >
> > > We're glad our rebuttal addresses your concerns and appreciate that you maintain your rating of 8.
> > >
> > > We will continue revising our paper based on constructive feedback from the reviewer and other reviewers. Please feel free to let us know if you have any further concerns.
> > >
> > > Best,
> > >
> > > Authors

---

### Meta-Review · Area_Chair_nqzQ · 2024-12-19

**Metareview:**

The paper introduces ACUS (Audio Captioning with Unbiased Sliced Wasserstein Kernel), a framework aimed at addressing key challenges in audio captioning, specifically exposure bias and temporal misalignment. The paper claims following contributions: 1) proposed unbiased sliced Wasserstein RBF (USW-RBF) Kernel, a novel kernel equipped with rotary positional embeddings for improved cross-modal similarity measurement between audio and textual data. The kernel effectively captures temporal information and addresses issues like dimensionality reduction and temporal distortion in traditional methods; 2) leverage stochastic decoding techniques (e.g., nucleus sampling and top-k sampling) during inference to reduce exposure bias; enhancing the diversity and quality of generated captions, aligning them better with audio inputs; 3) leverage contrastive learning to improve alignment between acoustic and linguistic features, reducing caption degeneration issues common in traditional training methods; 4) ablations and empirical results on benchmarks such as AudioCaps and Clotho showed the efficacy of proposed method.

Strength of this paper
- Tackles a relatively unexplored problem in audio captioning by leveraging cross-modal similarity calculations to capture temporal dynamics effectively. Proposed an interesting method of combining unbiased sliced Wasserstein RBF (USW-RBF) kernel with stochastic decoding methods for audio captioning and addressing common challenges such as exposure bias. The author also provides formal proofs for properties of proposed approach, including positive definiteness and unbiasedness, and demonstrates the convergence rate for Monte Carlo estimation. The work offers a fresh perspective in audio captioning by focusing on kernel-based similarity measures rather than proposing new architectures.
- Empirical results support authors claim and efficacy of proposed approach , where the ACUS framework improves caption diversity, length, and semantic alignment with audio events, generating more descriptive and meaningful captions validated through both quantitative and qualitative assessments. The framework is adaptable to a wide range of existing audio captioning models, enhancing their performance with minimal changes.


Weakness of this paper

Several reviewers raised few concerns/limitations of this paper. By addressing these limitations, the paper could strengthen its experiment and expand impact.
- Limited evaluations: key baselines, such as prefix-based architectures and other relevant approaches from literature, are not evaluated and compared with; the study is focused only on audio captioning tasks and does not explore other audio-text multimodal tasks (e.g., audio-text retrieval, automatic speech recognition) or benchmarks (e.g., SUPERB, Dynamic-SUPERB, or SpeechCaps) to demonstrate the framework's broader applicability. There is also limited discussion of the failure cases, model’s behavior and constraints. Some considers are also raised about the improvement , e.g., the SPIDEr scores improvement over baselines is minor.
- Computational overhead is another concern. Inference with stochastic decoding (e.g., generating multiple candidate captions) may lead to significant increases in computational time, potentially limiting its applicability in real-world, large-scale scenarios; there is no such analysis on the computational costs.
- The exposure bias is a general issue in auto-regressive networks and may not be as critical in this (AAC) domain. Thus the  motivation for addressing exposure bias in AAC tasks is not well-justified; in the meantime, alternative solutions, such as using reinforcement learning (RL) or larger decoders, might be even effective but are not discussed.

**Additional Comments On Reviewer Discussion:**

In addition to above weaknesses, reviewers also raised some other weaknesses (e.g., unclear/inconsistent writing and hard to understand, lack of details in integrating USW-RBF kernel and overall model architectures) and improvements during rebuttal. Although some of the weakness have been improved / somewhat addressed during rebuttal session (e.g., further explanation on the questions raised by reviewers, more discussion, clarification, and more experiment results added), overall review rating was not raised significantly, and the rating is still at borderline. I think the session is too short and some weaknesses are hard to address in such a short period of time. Also there is a general concern about the scope of the work, as AAC is a very niche area and proposed method might of limited utility. Given the high bar of ICLR, I think the paper is still of limited interests to the audience , and thus I recommend to reject the paper, and the authors to re-work on these weakness and re-submitting to future conferences.

---

### Decision · Program_Chairs · 2025-01-22

Reject